# Climate change induced complex shifts in snake distributions expose people to snakebite and threaten biodiversity

Anna F. V. Pintor[1,2]*, Kaushi S. T. Kanankege[1], Mike Turner[1,3], Bernadette Abela[1], Rafael Ruiz de Castañeda[1,4], Bethany Moos[1], Tamer A. Hasanein[5], Prashant Hedao[5], Kt Friar[5], Adam McKay[5], Gerardo Martín[6], Wolfgang Wüster[7], Romulus Whitaker[8,9], Fernando Martínez-Freiría[10], Kate Jackson[11,12], Laurent Chirio[13], Matthew LeBreton[14], Abubakr Mohammad Abdalhalee[15], Ulrich Kuch[16], Deb P. Pandey[17], Chifundera K. Zacharie[18], Maria Elena Barragan-Paladines[19], Carlos Yañez Arenas[20], Masoud Yousefi[21,22,23], Patrick K. Malonza[24], Mahmood Sasa[25], Zuhair S. Amr[26], Hammadi Achour[27], Anooshe Kafash[28], David J. Williams[29]

1 Department of Control of Neglected Tropical Diseases, World Health Organization, Geneva, Switzerland, 2 Australian Institute of Tropical Health and Medicine, James Cook University, Cairns, Queensland, Australia, 3 Division of Infection and Immunity, University of Glasgow, Glasgow, Scotland, United Kingdom, 4 Institute of Global Health & Division of Tropical and Humanitarian Medicine, Department of Health and Community Medicine, Faculty of Medicine, University of Geneva, Geneva, Switzerland, 5 GIS Centre for Health, Division of Data, Analytics and Delivery for Impact, World Health Organization, Geneva, Switzerland, 6 Departamento de Sistemas y Procesos Naturales, Escuela Nacional de Estudios Superiores unidad Mérida, Universidad Nacional Autónoma de México, Mérida, Yucatán, Mexico, 7 Molecular Ecology and Evolution at Bangor (MEEB), School of Environmental and Natural Sciences, Bangor University, Bangor, Wales, United Kingdom, 8 Centre for Herpetology, Madras Crocodile Bank, Chennai, Tamil Nadu, India, 9 Global Snakebite Initiative, Belmont, California, United States of America, 10 CRETUS - Department of Zoology, Genetics and Physical Anthropology, Universidade de Santiago de Compostela, Santiago de Compostela, Spain, 11 Whitman College, Walla Walla, Washington, United States of America, 12 Asclepius Snakebite Foundation, Aurora, Colorado, United States of America, 13 Koninklijk Belgisch Instituut voor Natuurwetenschappen, Brussels, Belgium, 14 Mosaic, Yaounde, Cameroon, 15 The Conflict and Environment Observatory, Hebden Bridge, West Yorkshire, United Kingdom, 16 Institute of Occupational, Social and Environmental Medicine, Goethe University Frankfurt, Frankfurt am Main, Germany, 17 Department of Veterinary Microbiology and Parasitology, Agriculture and Forestry University, Bharatpur Metropolitan City, Nepal, 18 Coordination of Research and Projects, Antivenom Center (CAV), University of Kinshasa, Kinshasa, Democratic Republic of the Congo, 19 Gustavo Orcés Herpetological Foundation, Quito, Pichincha, Ecuador, 20 Laboratorio de Ecología Geográfica, Unidad de Biología de la Conservación, Parque Científico y Tecnológico de Yucatán, Unidad Académica Sisal - Facultad de Ciencias, Universidad Nacional Autónoma de México, Mexico City, Mexico, 21 Leibniz Institute for the Analysis of Biodiversity Change (LIB), Museum Koenig, Bonn, Germany, 22 Department of Animal Science, School of Biology, Damghan University, Damghan, Iran, 23 Department of Biology, Faculty of Science, Hakim Sabzevari University, Sabzevar, Iran, 24 Section of Herpetology, National Museums of Kenya, Nairobi, Kenya, 25 Mahmood Sasa, Instituto Clodomiro Picado y Escuela de Biología/ Museo de Zoologia, Centro de Investigaciones en Biodiversidad y Ecología Tropical, Universidad de Costa Rica, San José, Costa Rica, 26 Department of Biology, Jordan University of Science & Technology, Ar-Ramtha, Jordan, 27 Institut Sylvo-pastoral de Tabarka, Laboratoire des ressources sylvo-pastorales, Université de Jendouba, Jendouba, Tunisia, 28 School of Culture and Society, Department of Archeology and Heritage Studies Research, Aarhus University, Aarhus, Denmark, 29 Prequalification Unit, Regulation and Prequalification Department, World Health Organization, Geneva, Switzerland

* AnnaFVP@gmail.com

## Abstract

Snakes play pivotal roles in many ecosystems. While some species, including medically important ones, are considered threatened by the IUCN, snakebite takes

**Data availability statement:** The data for this manuscript and a short user guide are stored in the Harvard Dataverse at https://doi.org/10.7910/DVN/QVWB4E. The data is also accessible through OPHIDS on the WHO Snakebite Information and Data Platform and updated versions will be uploaded when applicable [27].

**Funding:** This work was supported by Wellcome Trust (grant reference number 222215/Z/20/Z to DJW, MT, and AFVP). The funders had no role in study design, data collection and analysis, decision to publish, or preparation of the manuscript. AP received a salary as part of the funding received from Wellcome Trust (222215/Z/20/Z).

**Competing interests:** The authors have declared that no competing interests exist.

a heavy toll on rural agricultural populations in the developing world. Approximately 138,000 deaths and 400,000 disabilities result from snakebite annually and WHO has pledged to reduce the resulting health burden by 50% by 2030. Among a plethora of reasons for insufficient snakebite mitigation, one is limited explicit knowledge of how, where, and when humans and snakes interact, which limits the timely, accurate, and efficient deployment of resources. Here, we revise the list of medically important snakes based on recent taxonomic updates and use high-resolution data from a broad range of published and unpublished resources to compare expert-derived ranges with statistical geographical models of habitat suitability for all 508 most medically important snake species globally. Our study is the first to model every single medically important snake species including data deficient ones, at the highest resolution to date, and with the largest supporting occurrence dataset. We generate geographically explicit estimates of how much human and snake populations overlap (snake-human-overlap-index; SHOI), which is the most fundamental prerequisite for human-snake conflict to occur. Finally, we model the effects of climate change on snake distributions. We predict substantial, short- and long-term shifts in snake distributions, including range contractions for many threatened species and increased human exposure to species of major public health concern. In combination with other drivers of increased snake-human conflict, such as human behaviours and snake traits, our predictions can be used to decide where to stockpile which antivenom, how to ensure adequate capacity of individual health facilities, how to improve health care accessibility of remote at-risk communities, and where to focus conservation efforts for threatened snake species. Hence, we highlight the need for geographically targeted efforts to benefit both vulnerable human and snake populations, as part of a One-Health strategy.

## Author summary

Snakebite is a neglected tropical disease that affects millions every year and primarily results from conflicts in how humans and snakes use environments where they co-exist. Unfortunately, our understanding of where conflict is most pronounced is limited because of surprisingly sparse data on snakebite numbers, their locations, and snake distributions, especially in regions of the world where snakebites are most prevalent. Here, we collate unprecedented amounts of data on where medically important snake species occur and combine them with expert knowledge and statistical models of environmental suitability for snakes at a global 1km resolution now and under conditions of predicted climate change. This approach gives us detailed maps of snake-human population overlap, and allows stakeholders to implement targeted, future-proof efforts to improve human-snake co-existence as well as making snakebite treatments available in the most appropriate locations. It also enables us to detect knowledge gaps that

still need to be addressed, based on disagreements of expert knowledge and statistical models. Our results can be used as a basis to study factors that increase human susceptibility to snakebite in areas where they co-exist, such as agricultural practices, housing conditions, lifestyle traits of different snakes, or weather events.

## Introduction

Snakebite envenoming (snakebite hereafter) is a neglected tropical disease that affects millions, kills over 130,000 people and leaves 400,000 suffering from long-term physical and psychological medical conditions annually [1]. It is largely a disease of poor rural communities [2,3] in low and middle income countries (LMICs) [1,4] in tropical and sub-tropical regions. Consequently, the burden of snakebite has historically been inadequately addressed in global disease mitigation strategies [5]. Snakes also play pivotal roles in many ecosystems [6], yet their conservation status is concerning: over 30% of species in the most medically important venomous snake families (Viperidae and Elapidae) are threatened, near threatened or data deficient [7], many are geographically restricted, have declining populations [8], and are negatively affected by landscape conversion for human activities [9], illegal collection for the pet [10] or food trade [11], and indiscriminate culling out of safety concerns [12]. Unlike many infectious diseases, eradicating the disease-transmitting (snakes) or causing (venom) agent is not an option; venomous snakes and people must co-exist and share common ecosystems.

With the Global Strategy for Prevention and Control of Snakebite Envenoming [13], the World Health Organization (WHO) aims to reduce global snakebite deaths and disabilities by 50% by 2030. Understanding the geographic distribution of snakes, their overlap with human populations, and snakebite patterns across the world is, therefore, a crucial first step for both conservation and public health strategies and would support efficient positioning of mitigation efforts, such as risk-based placement of antivenoms, training healthcare staff to handle cases, and educating communities [14]. However, high-resolution data on these factors are sparse [15]. Inadequate resources, challenging field conditions, and complex political situations hinder data collection. Furthermore, future-proofing snakebite mitigation and conservation strategies involves assessing the likely impact of climate change on human-snake interactions [14], because the distributions of ectotherms such as snakes are highly dependent on climatic conditions [16].

WHO maintains a list of medically important venomous snakes (MIVS), categorized into two groups: 1. Primary MIVS that are highly venomous and common or widespread and cause many cases of bites, morbidity, disability or death; 2. Secondary MIVS that are highly venomous and capable of causing morbidity, disability or death, but either lack epidemiological data or bite less frequently because of their behavioural or ecological traits [17]. This list is the fundamental basis for attributing any spatial information to the correct species. However, the taxonomy of MIVS and information on the attributes that qualify them for listing are revised constantly, while updates of the MIVS list have been much less frequent.

Traditionally, species' distributions are recorded by collation of point occurrences and creation of expert derived range maps (EDRs) [18]. Recent advancement of online databases such as the Global Biodiversity information Facility (GBIF) [19], citizen science platforms such as iNaturalist [20] and extraction of location data from social media platforms (Facebook, Flicker etc.) [21] have greatly improved the accessibility of location information on MIVS. Despite these advances, data for many, even common species are still surprisingly sparse and even the most advanced studies on snake distributions only cover those MIVS with sufficient data [22]. Omission of data-sparse taxa negatively affects conservation outcomes [23], and snakebite mitigation. Additionally, the process of creating EDRs for field guides from these disparate data sources remains slow, repetitive, resource-intensive, and even contentious amongst experts.

More recently, Species Distribution Models (SDMs), and Environmental Niche Models (ENMs), have become widely used in ecology and conservation to estimate species' habitat suitability. They can offer detailed insights into poorly sampled species' distributions even from sparse occurrence data [24,25] combined with high resolution environmental data and provide much more fine-scale estimates of species' habitat suitability than coarsely 'drawn' EDRs [18]. Similar to EDRs, SDMs aim

to predict the realized distribution of species, while ENMs model conditions suitable for species to thrive and project them to those available across the landscape (only some of which are part of the realized distribution due to access restrictions or biotic interactions) [26]. ENMs can also predict future changes in habitat suitability based on climate projections. In combination, EDRs and ENMs can provide more robust information on species' distributions than either in isolation [18].

Here, we use a global, iterative, intensive data mining process across public and private databases, citizen science platforms, scientific literature, books, and social media to establish a constantly updated, dynamic database of all MIVS globally, their taxonomy, and their distributions. Resulting datasets include occurrence localities, Geographical Information Systems (GIS)-enabled EDRs and high-resolution ENMs, as well as estimates of human-snake overlap. ENMs and Snake-Human overlap estimates are available for current and future conditions (2050 and 2090). Data collation included rigorous vetting by an international expert panel comprised of leading researchers in the fields of snake distributions and taxonomy to guarantee a trustworthy one-stop-shop for researchers, governments, NGOs, and the general public.

The 'Occurrence Point Hub and Information Database for Snakes' (OPHIDS) is publicly accessible through the WHO Snakebite Information and Data Platform [27] and forms a fundamental basis for future studies on snakebite prevalence across different social, behavioural, economic, and environmental contexts.

Our main objective is to provide an overview of the potential contribution of one of several factors facilitating snakebite incidence, i.e., the overlap of human and snake populations, and provide a baseline consensus, dynamic dataset for researchers and governments to use for snake-human conflict mitigation. These data are of potential utility in guiding better adaptation strategies in public health and clinical management of snakebites by, for example, indicating which geographies need antivenoms supplies, as well as in conservation of ecologically important or rare snake species.

## Methods

For a more detailed methodology please refer to S1 Text. The authors welcome feedback on all parts of the methodology, to improve future versions of the database.

### Study species

We used the original listed WHO recognized MIVS from the 2018 version [1,17] and revised their taxonomy using published literature, and expert feedback. We included any currently recognized species that were either (i) already explicitly WHO listed [1], (ii) new species implicitly listed as part of a former parent species, or (iii) part of an unresolved species complex including listed species: we include 508 individual species in the updated list. Both, category 1 and 2 species were included. Because some species are considered different categories in different countries, it is impossible to separate the two groups. Many category 2 species may have significant impacts on people but are too poorly studied to quantify bite incidence. Including these species provides valuable information on potential underreporting or to identify knowledge gaps. See S1 Table for a complete species list. Note that many MIVS taxa are under ongoing revision and these will be reflected in future versions of the list of WHO recognized MIVS, which will be published iteratively online on the WHO Snakebite Information and Data Platform [27].

### Occurrence data

Occurrence records were collated from public, private, and citizen science databases (e.g., GBIF [19], iNaturalist [20]), museum records, books, a broad array of scientific literature, and verifiable personal observations from experts and social media (see S1 Text for more details). Descriptive localities were georeferenced, and their spatial uncertainty estimated using Google Maps [28] (S2 Text). Coordinates of records only displayed as maps in publications were extracted using DataThief [29] and map symbol radius was used as uncertainty.

All occurrence records were vetted for taxonomic and location accuracy by an expert panel of >30 experts from around the world. For ENMs, records were reduced to 75% of those with highest location accuracy, except for species

that were data deficient (< 20 records; the number of records below which model accuracy decreases most rapidly [30]) or in consistently poorly sampled regions, for which all available records were used. Where our minimum 'standard' of 20 unique records at 0.01 x 0.01 decimal degree resolution could not be reached, or where taxonomic or geographic delineation between closely related and ecologically similar species was poorly resolved, we ran multi-species models for 'modelling units' (MUs). These 314 MUs combine species only when necessary for modelling purposes, i.e., if they are data deficient or too difficult to delineate from each other due to insufficient taxonomic certainty in many areas, and they expected to be similar in their habitat requirements based on existing knowledge of their biology. For example, several data deficient species in the *Trimeresurus macrops* complex or *Gloydius strauchi* complex were modelled together. This approach was necessary to enable distribution estimates for often neglected range-restricted or poorly studied species. MUs function as 'umbrella species' by expanding predictions to include several ecologically similar and taxonomically closely related species, and, in most cases cover species groups that used to be considered one species. Because our data collection and vetting are iterative, future versions will aim to model these species separately as soon as sufficient knowledge to do so is available. However, excluding them here simply because of data restrictions could have substantially underestimated overall exposure of people to MIVS. Because neglected tropical diseases are a crisis discipline, providing the best available data for data deficient species and improving data as it becomes available is the best approach to prevent blind spots in our knowledge. MUs for all species are specified in the species list in S1 Table.

### Expert derived ranges

Previous WHO species' range estimates produced in 2018 [1] were converted to country specific polygons attributed with the correct risk category (1 or 2) for each snake in each country. Each species' polygons were adjusted to incorporate locations of cleaned, high confidence occurrence records from this study and any changes suggested and agreed on by the expert panel. Polygons were created for each species in each country and aligned with country borders according to vector layers provided by the WHO GIS centre for health. These same layers were also used for any maps presented in this study.

### Environmental data

We used environmental data layers from a broad range of sources, including data on climate, topography, vegetation, land use, soil, water availability, and human population density (see S1 Text and S3 Text for details). All datasets were sourced at 1km resolution or finer and, where necessary, converted to raster, summarized across time steps, and resampled or aggregated to 0.01 decimal degrees with Geographic Coordinate System World Geodetic System 1984 (GCS WGS84) projection and a global extent (S3 Text).

Candidate variables were assessed for collinearity and one of each pair with correlation coefficients >0.8 removed based on biological relevance [25]. Only two sets of collinear variables were maintained based on expert experience of complementarity of those variables in other analyses despite collinearity. Collinearity has been shown to be of minor importance to Maxent model performance [31], and our variable selection process eliminates any redundancy in final model predictors. Therefore, including some collinear variables was considered preferably to excluding potentially biological important variables. The resulting set of variables included Temperature Seasonality, Maximum Temperature, Minimum Temperature, Annual Precipitation, Precipitation Seasonality, Precipitation of Driest Quarter, Precipitation of Warmest Quarter, Minimum Radiation, Maximum Radiation, Minimum Relative Humidity, Topographic Ruggedness Index, Degree of Orientation Towards the Equator, Mean Dry Matter, Range of Dry Matter Productivity, Mean Fraction Photosynthetic Active Radiation, Range of Fraction Photosynthetic Active Radiation, Landcover Type in 2018, Topsoil Bulk Density, Fraction of Coarse Fragment, Percent Clay, Percent Organic Carbon, Soil Type, Distance to Permanent Water, and People per Grid Cell in 2020 (as estimate of human modification of landscape).

Climate conditions for two future time steps (2050 and 2090 centred) were sourced from WorldClim [32] (CMIP6 [33] pathway SSP5-8.5 'business as usual'). Future climate for relative humidity and radiation were not available from WorldClim and were instead calculated from change grids sourced from Copernicus [34]. Seven Global Climate Models (GCMs) were available from both WorldClim and Copernicus and were used for projections: (1) CanESM5-CanOE (Canada); (2) CMCC-ESM2 (Italy); (3) EC-Earth3-Veg-LR (Europe); (4) FIO-ESM-2–0 (China); (5) INM-CM4–8 (Russia); (6) INM-CM5–0 (Russia); (7) MPI-ESM1–2-LR (Germany). We only used one future pathway, often considered the 'business as usual' pathway, because of computational limitations projecting over 300 MUs to two future timesteps at our comparatively high resolution of 0.01 decimal degrees (~1km) across seven GCMs. Using SSP5-8.5 allows us to plan for the likely worst-case scenario. Real changes may be less severe in the long term (2090) if climate change prevention strategies are ultimately adapted more widely, but 'real' effects are likely to lie somewhere between current conditions and SP5-8.5 in the shorter term (2050). For example, Capellán-Pérez et al. estimate that there is a 92% chance we will surpass total radiative forcing considered in pathway RCP-4.5 (most similar to current pathway SSPS-4.5), one of the most commonly used 'intermediate' emission scenarios [35].

## Modelling methods

Habitat suitability of each MU was modelled using Maxent 3.4.4 [36]. Maxent has frequently been shown to outperform many other approaches to modelling habitat suitability [37]. Some studies have recently used ensemble approaches to estimate the uncertainty involved in the selection of modelling method; however, these approaches may also artificially introduce uncertainty by including less reliable methods [38,39]. Given the enormous computational requirements involved in modelling over 300 MUs and projecting them to 2 future time steps under 7 GCMs each at a very high resolution of ~1km, we limited our models to one method that is known to perform well.

To approximate the area which each MU could have reasonably had access to in their recent evolutionary history and to reduce the effect of spatial sampling bias [40], we used a 'target group' background [25,41] combining all occurrence data from this study and all snake records available in GBIF [19], cut down to a buffer around the occurrences of each MU with width equal to maximum range of the MU in either latitudinal or longitudinal direction. A minimum buffer size of 1000km and maximum of 3000km was allowed to avoid unrealistically small or large backgrounds unlikely to represent habitat available to a species in its recent biogeographic history [25,40]. Target group backgrounds have been shown to be control sampling bias effectively when using Maxent models, by providing comparison data of where similar species with similar detection abilities have been sampled [41]. Limiting background data to habitat available to a species in its recent biogeographic history reduces overestimation of their niche [40]. Since the actual available area is not known, we opted to adjust background areas to species' current range size as a proxy of their degree of generalism and dispersal ability. We opted for a relatively large background, because (1) overestimation of suitable habitat is less detrimental that underestimating where species may overlap with human populations in our case, and (2) to ensure enough background points even for restricted species, as relevant taxa are only sparsely sampled across many regions of the World. Final backgrounds points varied between MUs from 1,042–357,124 (see S1 Table).

Data from environmental layers was extracted for each MU's presence and background samples and used for 'starting' models with all variables. Maxent was run with 10-fold cross validation using 70% of subsampled training and 30% test data, without extrapolation, allowing for linear, quadratic, product, and hinge features. We did not allow for product features for species that were both, data sparse (<80 records [25]) and naturally restricted (background buffer <1000km), to avoid model overfitting. The default logistic model parameters were used instead of species-specific tuning (i.e., the default convergence threshold, regulation, and count of iterations) to ensure that methods were comparable among species. We opted not to thin occurrence records (beyond only using one point per 1km grid cell) because the benefit of thinning to reduce potential spatial autocorrelation is equivocal [42,43] and removing points for many of our data deficient species' was considered more detrimental than potential spatial autocorrelation, especially since our target background already accounted for spatial sampling bias.

A second set of 'reduced' cross-validated models using the same settings as before was run including only the best performing variables according to ranked average permutation importance in starting models, which has been shown to correlate with empirically measured biological relevance of variables [44]. The five best performing variables were always maintained to avoid oversimplification of niche requirements [45]. All other variables with permutation importance below 1% and variables ranked past 1/20 of the number of occurrence records were excluded to avoid model overfitting by balancing numbers of data points and predictors. These cross-validated models were used for model evaluation. Finally, a 'full' model using 100% of records for training was run and projected to the landscape at 0.01 decimal degrees in GCS WGS84 for current, 2050- and 2090-centered climate. Future models were summarized across the 7 GCMs by calculating the median (most likely), 90% quantile (maximum likely) and 10% quantile (minimum likely) future habitat suitability. Only climate was varied between current and future conditions and all other variables were kept constant. While some variables are not static in reality, they are often heavily influenced by complex, fine-scale human decision-making processes (e.g., population density, land use and vegetation changes) and unlike with climate models, future estimates for them would have to be based on singular models available from the literature (i.e., no uncertainty estimates would be possible). Hence, we opted to avoid making assumptions about their future patterns at this point.

## Model processing

Models were subjected to thresholding using the 'balance training omission, predicted area and threshold' value provided by Maxent and rasters were rescaled to 0–1 after thresholding to make outputs for different species with different threshold values comparable.

For each MU we calculated the cost distance from each known, reliable occurrence point, using habitat suitability as a 'cost raster' [46]. A cost raster assumes that every grid cell has a cost associated with traveling through it proportionate to its habitat suitability. The cost distance from known occurrences then functioned as a measure of how certain we are that any grid cell is occupied by the MU based on how much resistance needs to be overcome to reach it from known occurrence localities [46]. We excluded any areas that were >500 cost distance units away from known occurrences. This removed any disconnected suitable areas, such as land masses separated by water or other dispersal barriers, and marginal areas (lower suitability areas connected to, but far from known occurrences). Outputs were cut to within the model's target group background area.

Lastly, we separated habitat suitability models of any species that had been modelled jointly in an MU by assigning each grid cell to that species for which cost distance was lowest at that location, a process previously used to separate genetic lineages within species modelled together [46]. We allowed an overlap between species of 500 cost distance units (CDUs), faded out proportional to overlap ratio (e.g., if both of two overlapping species had CDUs of 250 in a cell, both had their suitability downweighed by 50% there, effectively reducing certainty of suitability in areas within 500 CDUs of more than just one species). The downweighed output was then re-thresholded with the original Maxent threshold, thereby eliminating any cells of low suitability too close to another species in the MU. This overlap and fade-out method was used to capture potential contact zones and allow for some uncertainty in the range delimitation between species that were modelled as one MU.

All models are available in the original version, as well as the versions restricted using CDUs.

We multiplied the suitability across the predicted distribution for each individual species with the natural logarithm of human population density [47] to get an estimate of how much snake populations and human populations overlap (Snake-Human Overlap Index; SHOI).

Summary rasters were created for closely related species groups or clades (from here on referred to 'species groups'; e.g., all African spiting cobras), using maximum value of habitat suitability for any species in the relevant group. Groups were compiled by combining all closely related species that are unlikely to truly overlap in their distributions, i.e., where

any apparent overlap zones are more likely to represent transition zones, meaning their suitability is not additive in contact zones [48]. In most cases we expect one of these species to dominate in each grid cell and the species for which habitat suitability is greatest in any grid cell is likely to be the dominant one.

We also created cumulative suitability and cumulative SHOI rasters showing the sum of all merged suitability rasters for all species groups and the sum of all SHOIs for all groups. Species richness and/or cumulative suitability have previously been used successfully as predictors of snakebite incidence, especially if further weighed by relevant species traits [49,50].

## Statistics

All statistical analyses were performed in R 4.1.2 [51]. GIS analyses and data processing were performed in R 4.1.2, ArcGIS Pro 3.1 [52]. Cost distance calculations were performed on the R package 'gdistance' [53]. Circular statistics were performed using the R package 'circular' [54].

Model performance was assessed using area under the curve (AUC) and partial receiver operating characteristic (ROC) [55]. Additionally, models were validated by comparing them to EDRs and assessing omission (false negative; model predicts unsuitable habitat within EDR), commission (false positive; model predicts suitable area outside EDR), and congruence (both models and EDR predict area as occupied).

For AUC, Maxent outputs were used. Partial ROC compares 1-omission error of sample points at different percentages of predicted area at different threshold values to 1-omission error of random points at the same percentages of predicted are, using the area under both curves. To calculate partial ROC we ran 39 iterations comparing the ROC of 50% sub-sampled occurrence points to the ROC of the same number of randomly selected points (null expectation).

Trends in species' range size, total suitability, total SHOI, and range shift vectors were summarized individually and by biogeographic region, i.e., groups of countries that share similar snake fauna (Australasia, Southeast Asia, South Asia, East Asia, Middle East and Central Asia, Europe and Russia, North America, South America, Central America, and Africa).

Individual species trends in distribution size, total habitat suitability, and SHOI with climate change were assessed by calculating the number of occupied grid cells (range size), number of newly occupied cells (range expansion) and newly unoccupied cells (range contraction), sum of all suitable cells (total suitability), and sum of SHOI in all occupied cells (total overlap) at each time step (2050 and 2090).

Direction of range shift vectors were calculated based on the current and future distribution centroid (mean of latitudinal and longitudinal coordinates of all suitable grid cells). Vector length (shift distance in meters) was calculated as

$$
\begin{aligned}
D1 <- R^* (2^* atan2(sqrt\ (sine\ ((Lat_{future} - Lat_{current})/2)\ *\ sine\ ((Lat_{future} - Lat_{current})/2) \\
+\ cosine\ (Lat_{current})\ *\ cosine\ (Lat_{future})\ *\ sine\ ((Lon_{future} - Lon_{current})/2) \\
*\ sine\ ((Lon_{future} - Lon_{current})/2)),\ \ sqrt(1 -\ (sine\ ((Lat_{future} - Lat_{current})/2) \\
*\ sine\ ((Lat_{future} - Lat_{current})/2)\ +\ cosine\ (Lat_{future})\ *\ cosine\ (Lat_{future}) \\
*\ sine\ ((Lon_{future} - Lon_{current})/2)\ *\ sine\ ((Lon_{future} - Lon_{current})/2)))))
\end{aligned}
\tag{1}
$$

Where D1 is distance in meters, R is the radius of the Earth in meters (6,371,000), and Lat and Lon are the latitudinal and longitudinal position of points on a sphere in radians.

Vector direction was calculated as

$$
\begin{aligned}
D2 = (180/pi)\ *\ atan2(sine\ (Lon_{future} - Lon_{current})\ *\ cosine\ (Lat_{future})\ ,\ cosine\ (Lat_{current}) \\
*\ sine\ (Lat_{future})\ -\ sine\ (Lat_{current})\ *\ cosine\ (Lat_{future})\ *\ cosine\ (Lon_{future} - Lon_{current}))
\end{aligned}
\tag{2}
$$

Where D2 is the circular direction in degrees.

## Results

### Model performance

Most of the 314 MU's final ENMs performed well according to average test AUC > 0.85 across ten replicates, despite a high percentage of data reserved for testing (30%; Fig 1A). Of the ten models with poorer performance, those for five MUs (*Dispholidus typus*, *Pseudonaja textilis*, *Bothrops alternatus*, *Acanthophis antarcticus*, and *Pseudechis porphyriacus*) belong to wide-ranging species with many occurrence records (>700) and are representative of an overall decrease in both training and test AUC towards models with larger numbers of points, typical for ENMs [30]. The other five (*Calliophis philippina/bilineata/salitan/suluensis*, *Atractaspis bibronii/duerdeni*, *Eristicophis macmahonii*, *Micrurus sangilensis/clarki*, *Atheris broadleyi/subocularis*) were MUs that include data deficient taxa and had test AUCs of 0.777 to 0.846, substantially below their respective final training AUCs of 0.849 to 0.987, i.e., had good final model fit but poor representation of test conditions by training conditions.

9 models did not predict presences better than random based on partial ROC ratio (Fig 1B). Four were data poor models (≤20 records <; *Proatheris superciliaris*, *Bungarus bungaroides/slowinskii*, *Bitis parviocula/harenna*, and *Eristicophis macmahonii*), one was the most wide-spread of all venomous snakes (*Vipera berus*), three were restricted island or peninsular species (*Bothrops lanceolatus*, *Protobothrops flavoviridis*, and *Cortalus enyo*), and one, *Vipera monticola*, had large inaccessible areas predicted as highly suitable.

ENMs predicted substantial additional suitability outside EDRs for species with few occurrence records but less so for well-sampled, wide-ranging species (Fig 2A; Spearman's ρ = -0.46). Accordingly, congruence between ENMs and EDRs increased with more occurrence records (Fig 2B; Spearman's ρ = +0.53). More well-sampled species had a higher percentage of their total predicted suitable habitat within EDRs. The percentage of EDR area predicted as unsuitable (i.e., EDR omission) was low (median = 5.14%) independent of number of occurrences (Fig 2C; Spearman's ρ = +0.06). In summary, models overpredicted species' ranges more often than they underpredicted them. We have been careful to take account of possible over- and underpredictions in all interpretations below.

Fig 3 shows an example of a wide-ranging African species with varying degrees of EDR under- and overpredictions across its range, the black mamba (*Dendroaspis polylepis*). Underpredictions were common where anecdotal expert knowledge suggests occasional presences without accurate location information (e.g., *D. polylepis* habitat in Nigeria or Sudan). Often suitability within EDRs was patchy, leading to apparent underpredictions. Overpredictions often included marginal or poorly sampled habitat adjoining EDRs (e.g., poorly sampled regions in Eastern Namibia and Western Botswana for *D. polylepis*). The average suitability per grid cell for each species outside its EDRs was, on average, much lower than within EDRs (0.21 compared to 0.46, paired Wilcoxon signed rank test; V = 113136; p-value = $2.2*10^{-16}$).

### Climate change impacts on distributions and SHOI

Some species were predicted to experience major range contractions, and progressive decreases in suitability with climate change (e.g., Puff adders, *Bitis arietans*), while some species' ranges were predicted to remain relatively stable (e.g., East African carpet vipers, *Echis pyramidum*), and some were predicted to experience large increases in suitability and range expansions (e.g., Black-necked spitting cobras, *Naja nigricollis*; Fig 4).

Median predicted increases in range size correlated positively with predicted increases in total habitat suitability (sum of suitability of all grid cells; linear regression; $F_{1,507} = 3926.5$ and p < 0.0001 for median change by 2050; $F_{1,507} = 6772.5$ and p < 0.0001 for median change by 2090), and with increases in overlap of snake and human populations ($F_{1,507} = 431.7$ and p < 0.0001 for median change by 2050; $F_{1,507} = 302.2$ and p < 0.0001 for median change by 2090). The four species with highest predicted 'confident' positive change in SHOI (i.e., where the predicted change in 2090 SHOI was positive even for 10% quantile predictions) were the black-necked spitting cobra (*Naja nigricollis*), many-banded krait (*Bungarus multicinctus*), cottonmouth (*Agkistrodon piscivorus*), and copperhead (*Agkistrodon contortrix*; Fig 5).

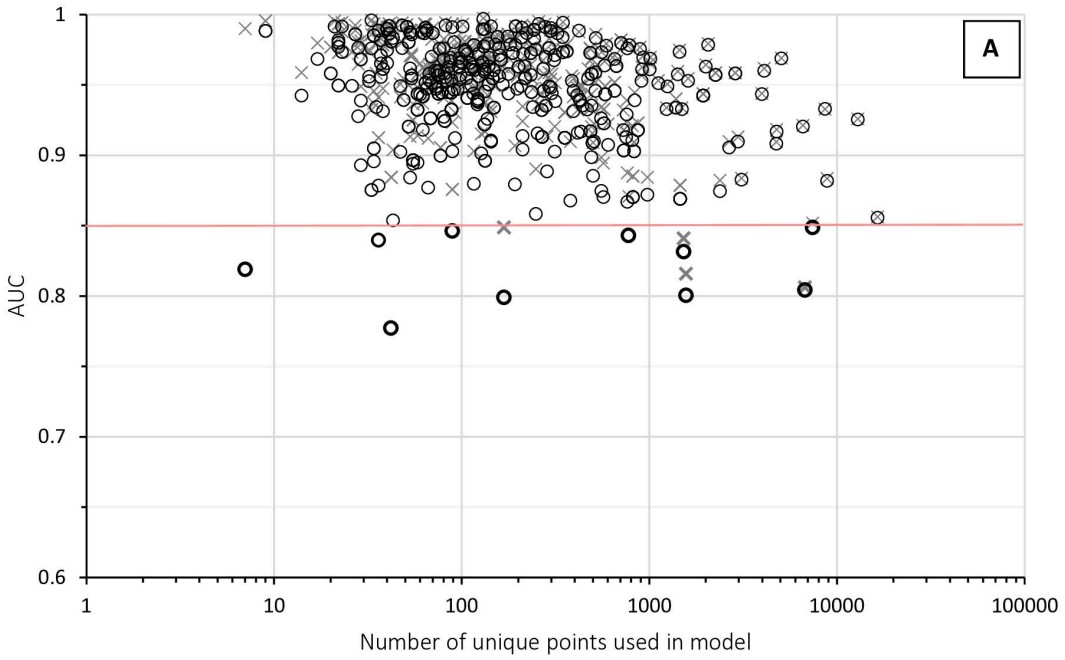

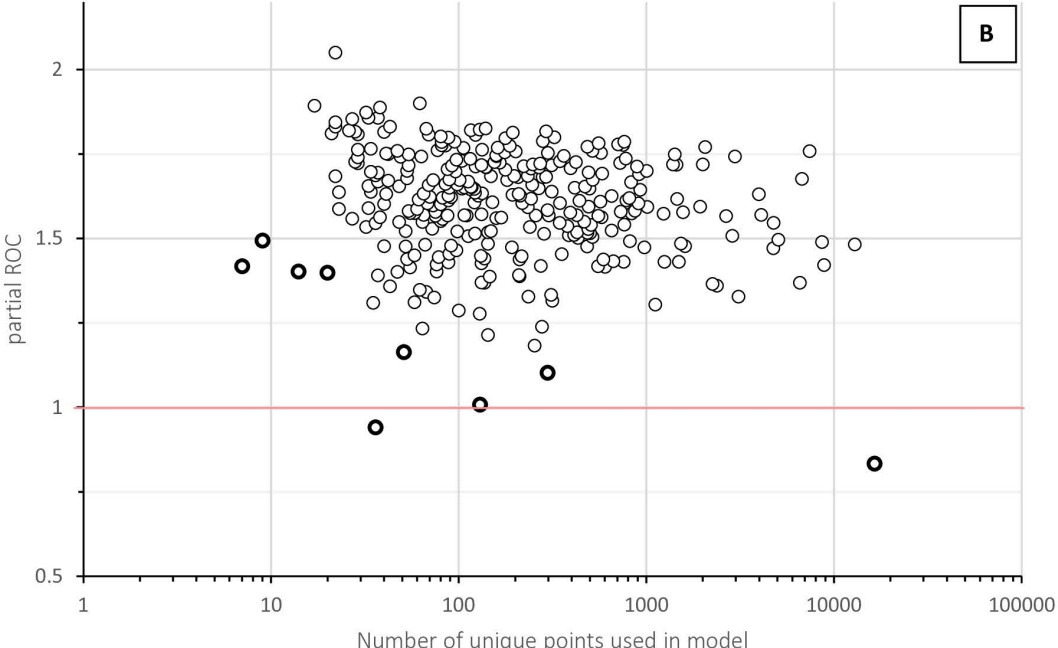

**Fig 1. Model validation using AUC and partial ROC.** AUC and partial ROC vary across models with different number of underlying occurrence records: (A) Test AUCs of 10-fold, 30:70% subsampled, cross-validated Maxent models (black circles) show how well models trained on 70% of the data predicted the 30% of data set aside for testing; final model training AUCs (grey crosses) show the performance of the final models using 100% of the data. The models with performance below an AUC of 0.85 are indicated in bold. (B) Average partial ROC ratio shows which models are significantly different from 1 (black circles; better prediction than by chance) and which are not (bold circles).

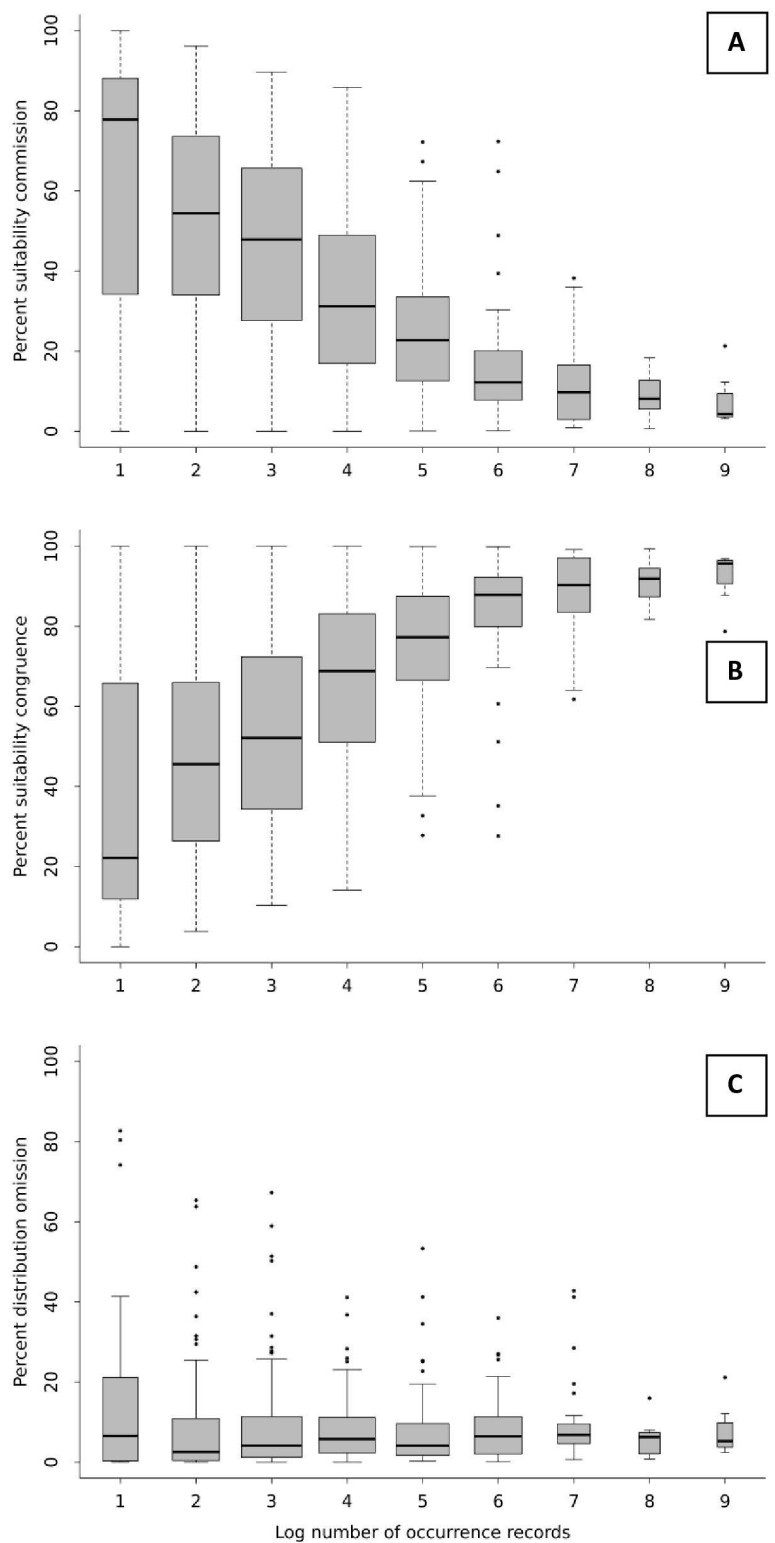

**Fig 2. Trends in model commission, congruence, and omission with across number of points used in models.** Boxplots showing how model commission (A), congruence (B), and omission (C) varied across number of occurrence records used for models. Commission and congruence were measured as percentage of total sum of cells predicted as suitable: high commission (false positives) means a high percentage of cells predicted as

suitable were outside the EDR for the species, i.e., models predicted additional suitable habitat to what experts believe, while high congruence (true positives) means a high percentage of suitable cells were within the EDR, i.e., agreed with expert opinion. Omission (false negatives) was measured as percentage of EDR predicted as unsuitable by statistical models, i.e., areas where experts believe the species to occur, but models did not confirm it.

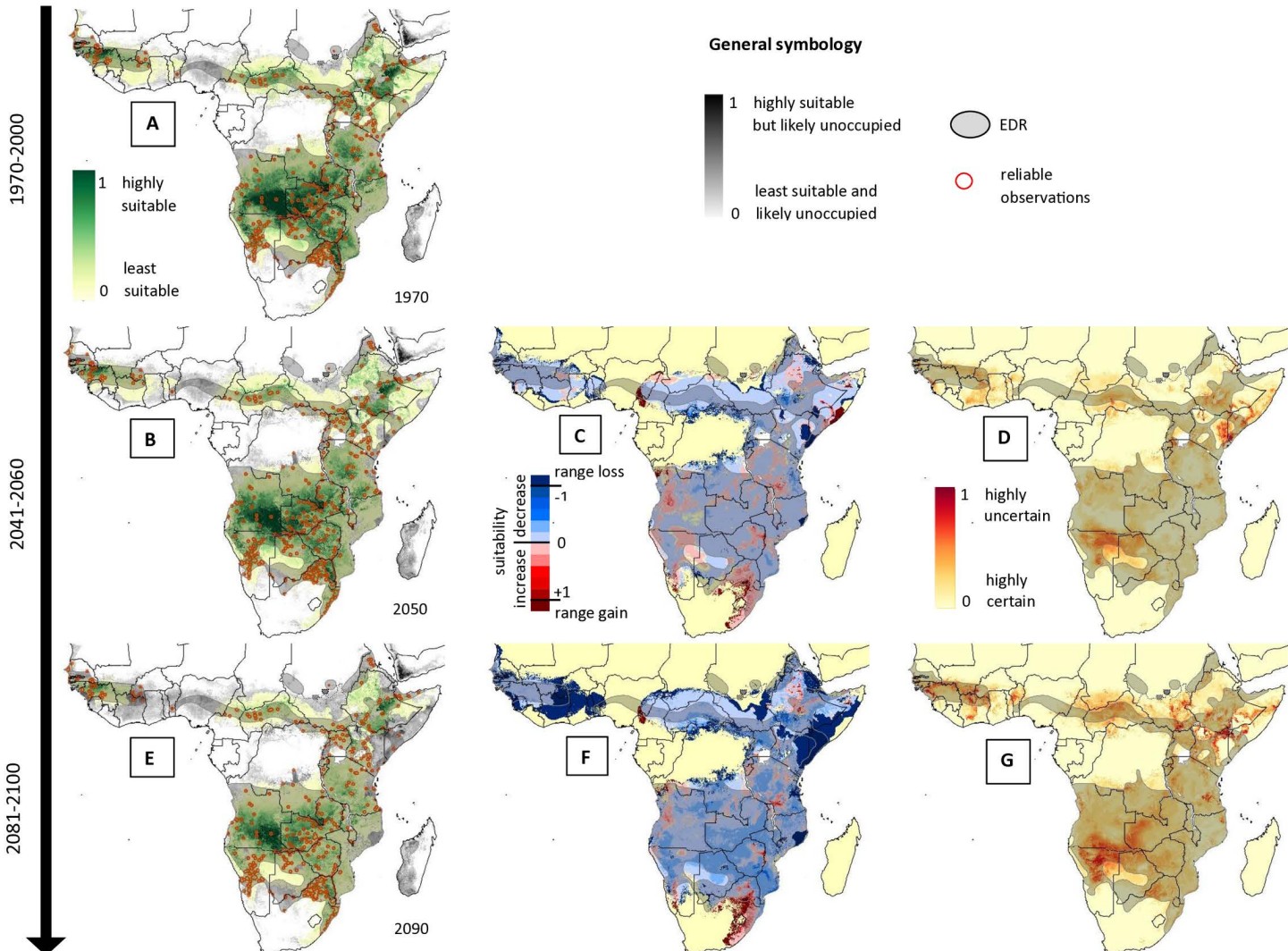

**Fig 3. Predicted changes in habitat suitability for the Black Mamba.** Current (A), 2050 (B-D), and 2090 (E-G) predicted habitat suitability (A, B, and E), change in habitat suitability from current (C and F), and uncertainty in prediction (D and G) for an example species, the Black Mamba, *Dendroaspis polylepis*. Predictions are overlayed with the current EDR for the species as shaded blue area and with recorded reliable observations as yellow points. Grey shaded grid cells in the background show the raw predicted habitat suitability before areas at high cost-distance to known occurrences were cut out. These areas are likely unoccupied due to patchy connection to main habitat or high cost-distance to known areas of occurrence (light grey to dark grey) but are technically suitable and may be shown to be occupied pending further targeted data collection. Maps were created in ESRI ArcPro 3.1.0 [52]. Basemap shows WHO admin 0 country boundaries 2024 (CC BY 4.0). The designations employed and the presentation of the material in this publication do not imply the expression of any opinion whatsoever on the part of WHO concerning the legal status of any country, territory, city or area or of is authorities, or concerning the delimitation of its frontiers or boundaries.

 **Neglected Tropical Diseases**

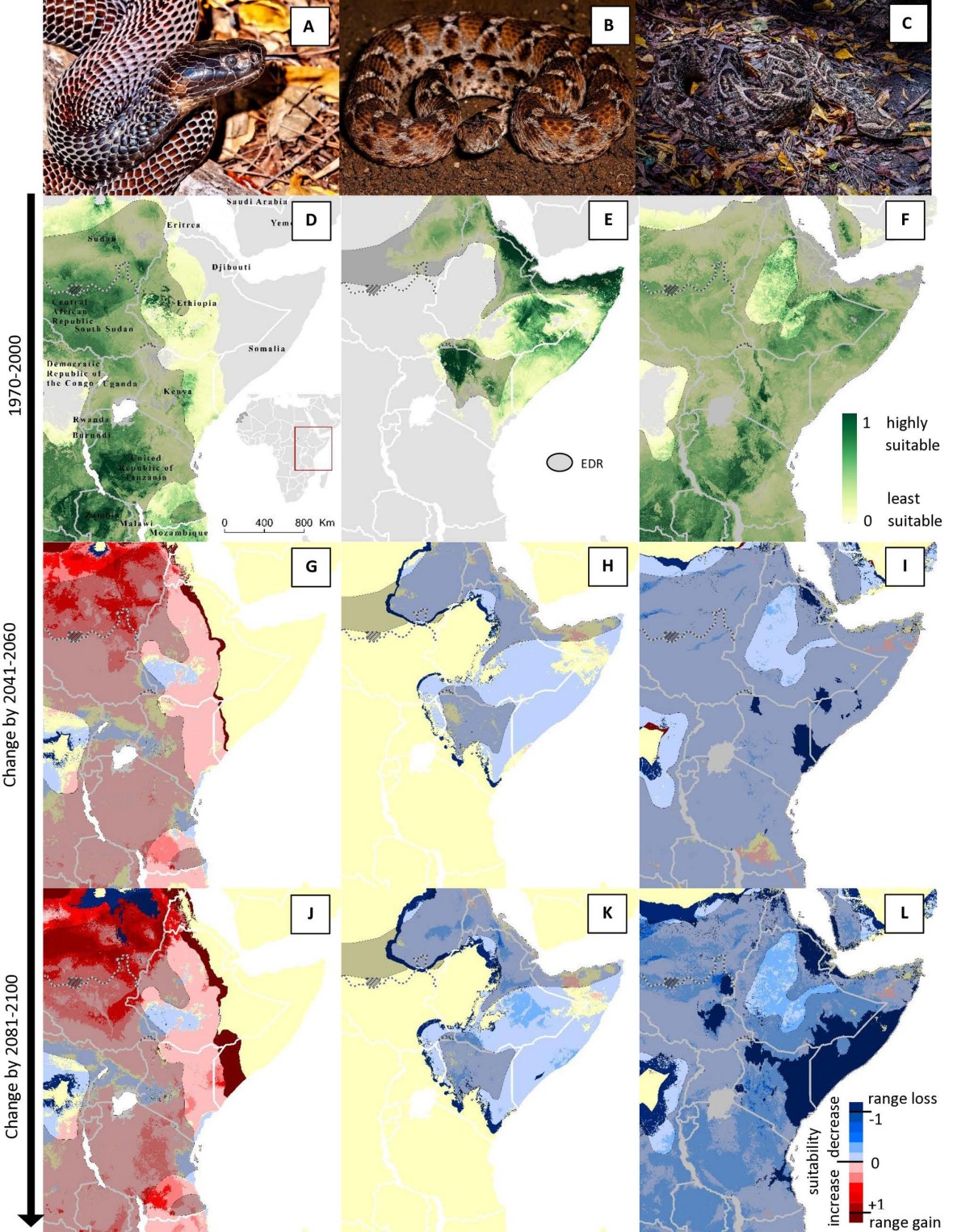

**Fig 4. Examples of different responses to climate change predicted for different snakes.** Examples of current habitat suitability (D-F) and predicted changes in suitability by 2050 (G-I), and 2090 (J-L) for three example species (photos of species shown in A-C) of high medical importance. Overall suitability is predicted to increase greatly for the Black-necked Spitting Cobra, *Naja nigricollis* across East Africa (A, D, G, J), remain relatively

stable for the Northeast African Carpet Viper, *Echis pyramidum* (B, E, H, K), and decrease for the Puff Adder, *Bitis arietans* (C; F; I; L). Maps were created in ESRI ArcPro 3.1.0 [52]. Basemap shows WHO admin 0 country boundaries 2024 (CC BY 4.0). The designations employed and the presentation of the material in this publication do not imply the expression of any opinion whatsoever on the part of WHO concerning the legal status of any country, territory, city or area or of is authorities, or concerning the delimitation of its frontiers or boundaries.

For *Naja nigricollis*, SHOI increases were distributed across the whole range of the species, while for the other three they were mostly expressed as shifts in SHOI towards higher latitudes.

Notably, most species did not show a monotonous change in range size, suitability, and SHOI but rather a combination of range contractions and suitability decreases in parts of their range, concurrent with range expansions and suitability increases in other parts of their range (e.g., *Naja nigricollis*; Fig 4), leading to an overall shift in occupied area. Overall, there was a median trend for species from each biogeographic region to shift towards higher latitudes, despite substantial variation in individual species' shift directions within regions (Fig 6). For example, individual Australian species were predicted to shift towards the coast, whether this is North or South, although the median regional trend was towards South.

**Patterns across regions**

Across all species globally, current SHOI was predicted to be highest across much of sub-Saharan Africa, as well as South and Southeast Asia. Patchier but substantial areas of SHOI were seen across Central and South America, and the Middle East (Fig 7). Future SHOI increases were most notable across the USA, the People's Republic of China, the Indian subcontinent, parts of Western Africa (Fig 7E and 7H), northern Angola, countries bordering the Eastern Congo Basin, and the northern parts of the Andes in South America (Fig 7 F and 7I).

Species richness for MIVS (Fig 8A, 8B and 8C) was highest across the Amazon basin, the Congo basin and surrounding countries, and Southeast Asia. Species' long term range expansions (Fig 8D, 8E and 8F) were most pronounced in China and the USA towards the North, in Australia towards the South, in countries bordering the Congo basin, in northern West Africa (Fig 8E), towards higher elevations in central and northern South America (Fig 8F), and in Northeastern South Africa. Range contractions (Fig 8G to 8I), on the other hand, were more pronounced worldwide and particularly across the Congo basins, coastal West Africa (Fig 8H), most of lower elevations in South America (especially across the Amazon basin), across Southeast Asia, and in central and northern Australia.

When looking at the species with highest overall SHOI per biogeographic region, some regions showed a clear dominance of a few species (Australasia, Europe and Russia, and North America; Fig 9), while others had a richer array of species contributing to overall SHOI, with 'other' species contributing more than the highest individual SHOI-causing species (Africa, Middle East and Central Asia, South, Southeast and East Asia, Central and South America; Fig 9). In many regions, the species with highest SHOI were in agreement with species known to contribute significantly to the burden of snakebite in those regions.

The ten medically relevant snake species with the highest current cumulative SHOI (y axis, shown in millions) across each biogeographic region, and for all other species present in each region combined (last bar). Double lines indicated interrupted bars for 'other' values much higher than the primary species (actual values printed on bar). For region boundaries see Fig 6.

Most species from each region show a median decrease in predicted range size, total suitability, and SHOI by 2090 (Fig 10), many by more than 50%. However, a substantial proportion of species in some regions also show predicted increases in all three metrics, most notably in North America, Central America, Africa, and East Asia.

Fraction of species in each region with predicted decreases (light blue equals 5–50% and dark blue >50% decrease), increases (pink equals 5–50% and red > 50% increase), and negligible changes (light grey; < 5% increase or decrease) in range size, suitability, and SHOI for 2050 and 2090. Fifty percent, i.e., half of all species present, is shown as grey line for

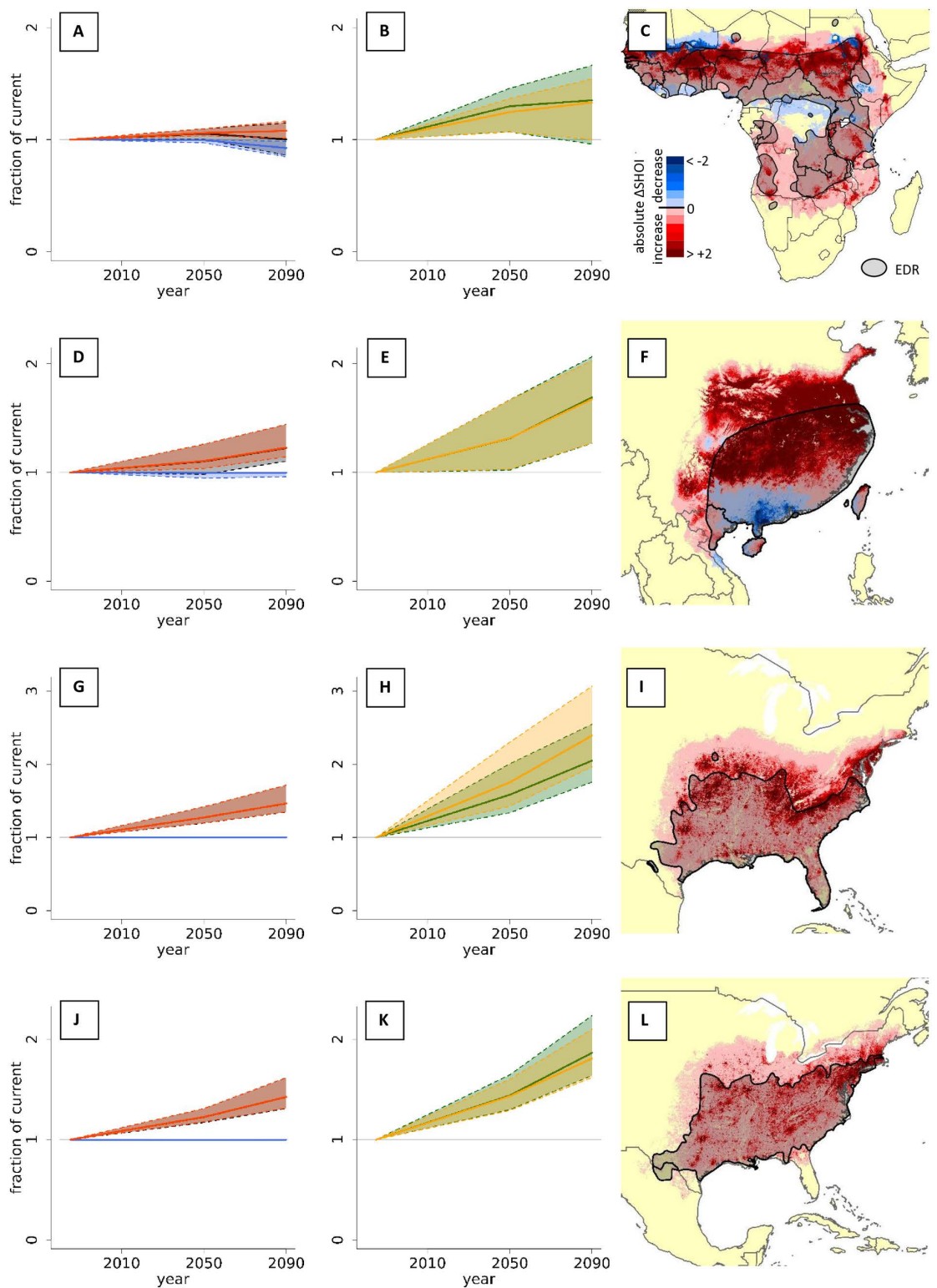

**Fig 5. Distribution shifts and SHOI changes in the snake species with highest predicted increase in overlap with people by 2090.** Trends in habitat suitability and SHOI for the four species with the highest significant (i.e., positive and different from 1) predicted increase in overlap with humans by 2090: *Naja nigricollis* (A-C), *Bungarus multicinctus* (D-F), *Agkistrodon piscivorus* (G-I), and *Agkistrodon contortrix* (J-L). A, D, G, and J show the

number of new grid cells that are predicted to become suitable (range expansion; red), and the number of cells that will stop being suitable (contractions; blue). The change in total number of suitable grid cells (grey), may be identical to range expansion in species where no contractions occur (G and J), remain constant for species that show similar amounts of expansions and contractions (A) or show an intermediate response (D). Despite some of the species experiencing both expansions and contractions, all show positive trends in total cumulative suitability (green) and SHOI (yellow; B, E, H, and K), either because increased suitability in their existing range or because of range shifts towards more populated areas.) C, F, I, and L show maps of predicted exposure decreases and increases across each species distribution. Maps were created in ESRI ArcPro 3.1.0 [52]. Basemap shows WHO admin 0 country boundaries 2024 (CC BY 4.0). The designations employed and the presentation of the material in this publication do not imply the expression of any opinion whatsoever on the part of WHO concerning the legal status of any country, territory, city or area or of is authorities, or concerning the delimitation of its frontiers or boundaries.

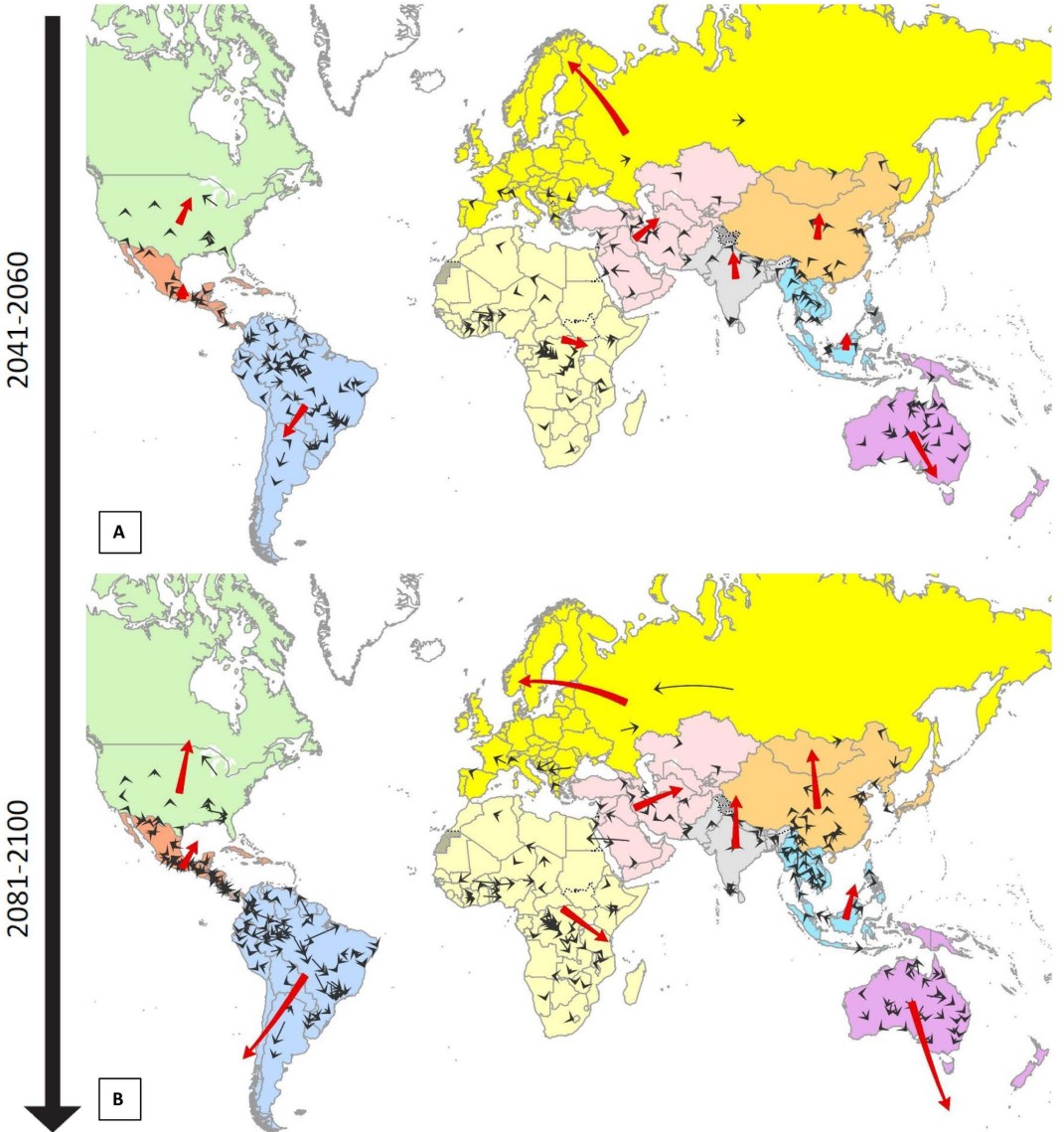

**Fig 6. Direction of predicted shifts in distributions for venomous snakes around the World.** Predicted range shifts for each individual species (small black arrows) as well as median (large red arrows) of all species' predicted range shifts for each biogeographic region (black outlines) for 2050 (A) and 2090 (B). Regional summary arrows are multiplied by factor 20 for better visibility. Maps were created in ESRI ArcPro 3.1.0 [52]. Basemap shows WHO admin 0 country boundaries 2024 (CC BY 4.0). The designations employed and the presentation of the material in this publication do not imply the expression of any opinion whatsoever on the part of WHO concerning the legal status of any country, territory, city or area or of is authorities, or concerning the delimitation of its frontiers or boundaries.

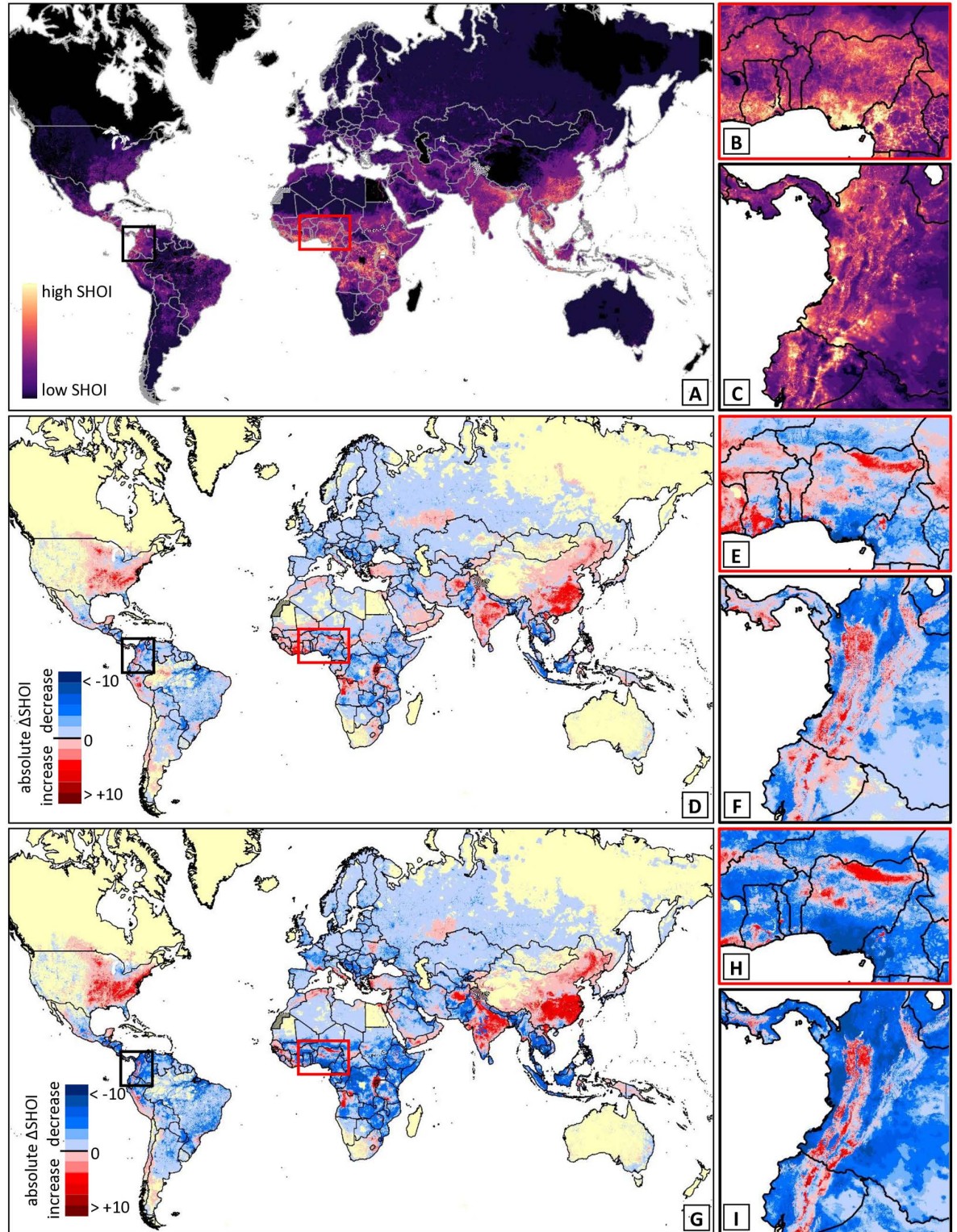

**Fig 7. Predicted changes in cumulative SHOI globally and in example regions.** Maps of total global SHOI (product of cumulative habitat suitability and log transformed human population density; A) and median predicted changes in total SHOI by 2050 (D) and 2090 (G). Insets show zoomed in maps for example regions in West Africa (red border; B, E, H) and North-Eastern South America (black border; C, F; I). The most widespread predicted

increases in overlap between medically important snakes and people can be seen in Eastern North America, across the Indian subcontinent, and Eastern Asia. Decreases are much more widespread. Within example regions, increases and decreases are patchier, with increases predicted along the Andes, Northern, and Western West Africa. Basemap shows WHO admin 0 country boundaries 2024 (CC BY 4.0). Maps were created in ESRI ArcPro 3.1.0 [52].

visual guidance. For most regions more than 50% of species show a decrease in all three metrics (either light blue or dark blue). For region boundaries see Fig 6.

## Discussion

Although recent studies have advanced our understanding of venomous snake distributions for certain taxa [56], regions [50,57], or even globally [22], this study is the first and most comprehensive to model environmental niches and predict distributions of all revised 508 MIVS of category 1 or 2 [1] (S1 Table) and their overlap with human populations. Such maps represent an unparalleled opportunity to spatially focus and future-proof snakebite mitigation strategies as well as conservation efforts.

The geographic distribution of MIVS and snakebite have mostly been studied at regional levels, at relatively coarse resolutions, or only for relatively data rich subsets of snakes [14,22]. The most comprehensive study on the topic to date modelled 209 MIVS' distributions [22]. We improve on this study substantially: instead of only using GBIF records, we undertook data mining of the literature and unpublished datasets to markedly improve datasets for all, but especially for data-deficient taxa, which enabled us to include all MIVS. For example, *Naja haje* and *Atheris broadleyi* in our dataset only included 46 and 2 unique records from GBIF, respectively, but 446 and 78 records from other sources. We revised the taxonomy of all MIVS and cover 508 species, with only 5 MUs (7 species) not reaching our desired minimum number of >20 occurrence records. For comparison, Martinez et al. 2024 used an older version of the MIVS list including 251 species, out of which only 209 met their data requirements. This is important because many taxa that have undergone substantial revisions recently. For example, Martinez et al. 2024 included *Naja melanoleuca* (now five species [58]). We established a panel of experts to vet all data and outputs, lending additional reliability to our results. We projected results to two different future time steps, used a 1km instead of 5km resolution and built models using a broad range of environmental variables instead of only climate. Most importantly all our MIVS information is continuously updated and made available online at regular intervals.

Our comparison of EDRs with statistical predictions of suitable habitats generated many examples of very strong overlap, but also some interesting anomalies that may guide where future field studies should focus on specific locations [59]. For example, *Echis leucogaster* and *Walterinnesia morgani* have both recently been found in areas disjunct from their previously known range that were predicted as suitable by our models before restricting them using CDUs [60] (David Williams pers. com.). Since we provide both, CDU-restricted and raw outputs, these can be used to provide higher confidence range estimates (CDU restricted) as well as other, potentially suitable habitat where further sampling would be beneficial (raw ENMs).

Notably, our analysis suggests that, at macro scales, SHOI (but not necessarily species richness) is overall highest in regions that are known to be high-burden areas for snakebite [1], such as South Asia and Sub-Saharan Africa. However, there is substantial variability in SHOI at smaller scales (one square km to the next) that is only detectable in high resolution analyses. Conditions that form the basis for high numbers of snakebites appear to result at least partially from both large human and snake populations in close proximity. Areas where SHOI is likely to increase include the Indian subcontinent, where snakebite burden is already very high [1,3], as well as regions that currently do not have high reported snakebite burden but could in the future because of high human population densities and predicted increases in snake habitat suitability, such as People's Republic of China and Eastern North America. Across Africa, South America, and Southeast Asia predicted changes in SHOI are less uniform, showing increases as well as decreases across neighboring habitat

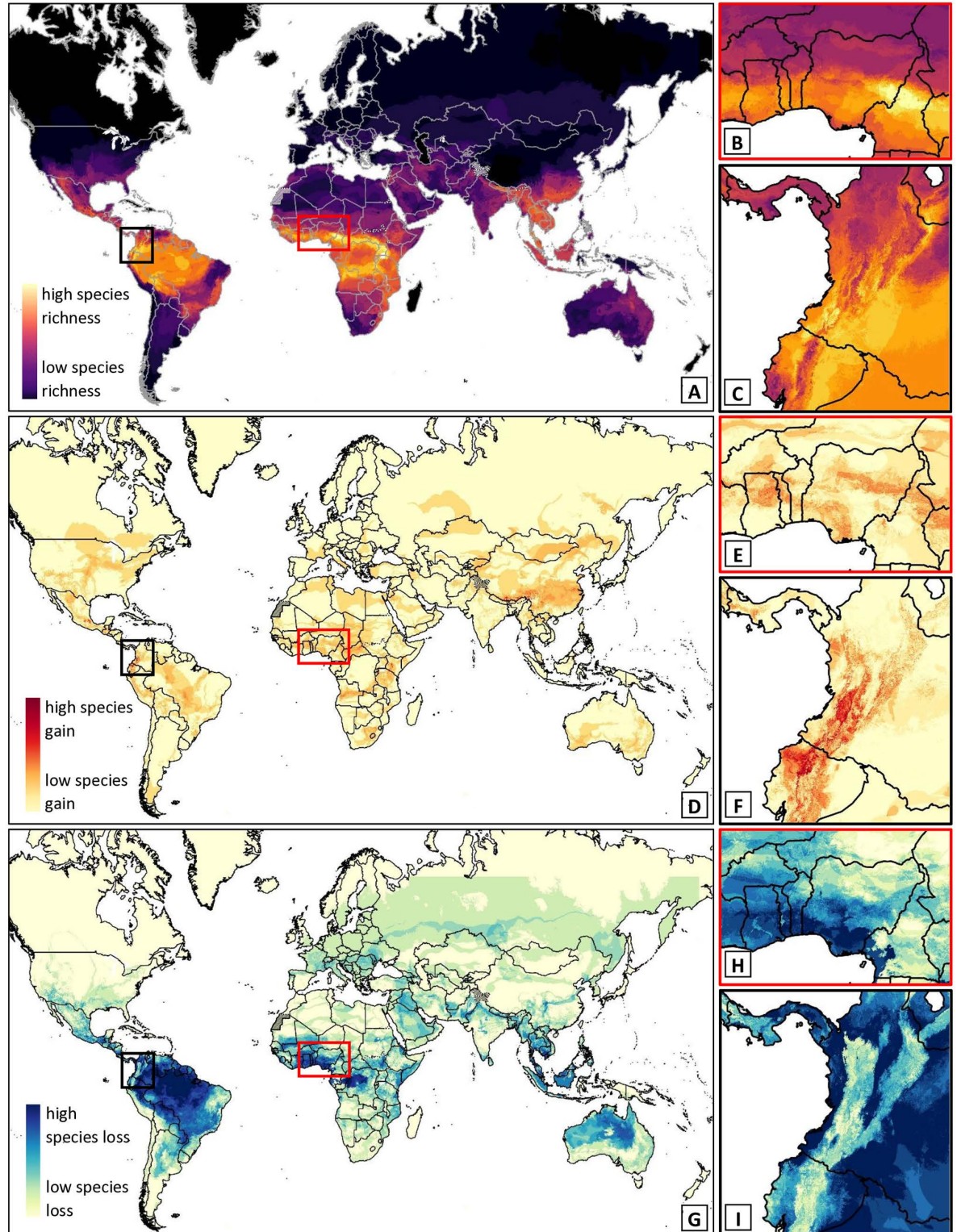

**Fig 8. Predicted changes in species richness globally and in example regions.** Maps of current global species richness for all snakes of medical importance (A) and predicted number of individual species gained (D) or lost (G) in each grid cell by 2090 according to median model projections. Insets show zoomed in maps for example regions West Africa (red border; B, E, H) and North-Eastern South America (black border; C, F; I) and mostly

patches within the landscape. Areas of increased future SHOI need investment into prevention strategies before shifts in snake distributions occur; some of these will apply to larger areas, while others will have to address differential changes at micro-scales. Since we did not vary human population density between current and future scenarios in our analyses, changes in SHOI are most likely underestimates.

We predict that snake distributions will generally shift towards higher latitudes in most regions (Fig 6), which is consistent with patterns observed historically for a wide range of taxa [61] and in previous predictive studies for snakes [50,57]. However, individual species' range shifts appear to be much more variable [62]. For example, in Australia most species show a coastward distribution shift in all directions away from the arid center of the continent, combined with a Southward shift of species along the East Coast; both trends are towards areas more highly populated by humans.

Our maps show geographic variability in SHOI at finer detail than most previous studies [14]. However, the overlap of human and snake distributions is only one factor determining risk of exposure to snakebites. Snake species vary considerably in their niche utilizations [63] and behavioural ecology [49], which will markedly affect how frequently they interact with humans. Also, species vary in their defensive behaviours [49]. Some snakes will take evasive action when disturbed and only bite as a last resort, whilst others are more likely to bite first, and then evade. Some snakes may also occur in much higher densities in their ideal habitat than others or be harder to avoid by people). Some of these trait differences are reflected in category 1 versus 2 listings for each species. However, each species category can vary among countries and often reflect lack of knowledge on their impact or differences in abundance or percent of the country covered by their distribution rather than true differences in their impact on people where they do occur. Using these categories to weight species impact is, therefore, not necessarily appropriate and much more research is needed to make it possible to include snake traits in risk analyses.

Human behaviour also affects snakebite risk [64]. Many people co-exist with snakes without much exposure to bites because their daily activities do not predispose them to encounter snakes. For example, agricultural workers with no access to protective equipment are much more exposed to contact with snakes than those using advanced machinery; workers operating in warm, wet seasons when snakes are more active are more likely to get bitten than others [64]. Our results, therefore, present a baseline on which further risk factors need to be assessed and should not be treated as an index of snakebite risk per se. Snakebite risk is the product of co-existence with snakes (SHOI) and other potential risk factors such as certain agricultural activities, lack of protective equipment [65], lack of snake-proof human and livestock habitations [66], occurrence of extreme weather events that increase snake activity or displace snakes and people [67], and cultural practices [68]. SHOI, however, is the most fundamental factor on top of which such other variables act.

Our distribution models showed high performance according to test AUCs despite large percentages of points set aside for testing (30%), good predictive ability according to partial ROC ratios, and good congruence with EDRs, especially for species with more occurrence records and larger ranges (Fig 2). Very few models performed poorly, usually for data-poor taxa. Only 5 MUs failed to meet the recommended minimum number of more than 20 unique records despite our best efforts to group data poor taxa into adequate MUs and 4 of them were among the poorly performing models. For example, *Eristicophis macmahonii* was the only species that performed poorly according to both AUC and partial ROC. This species is the most data poor of all our taxa (only 7 unique records) and had no closely related species it could be grouped with for modelling purposes. Several of the other poorly performing models were very wide-ranging taxa. It is harder to distinguish between suitable and unsuitable areas if an organism is

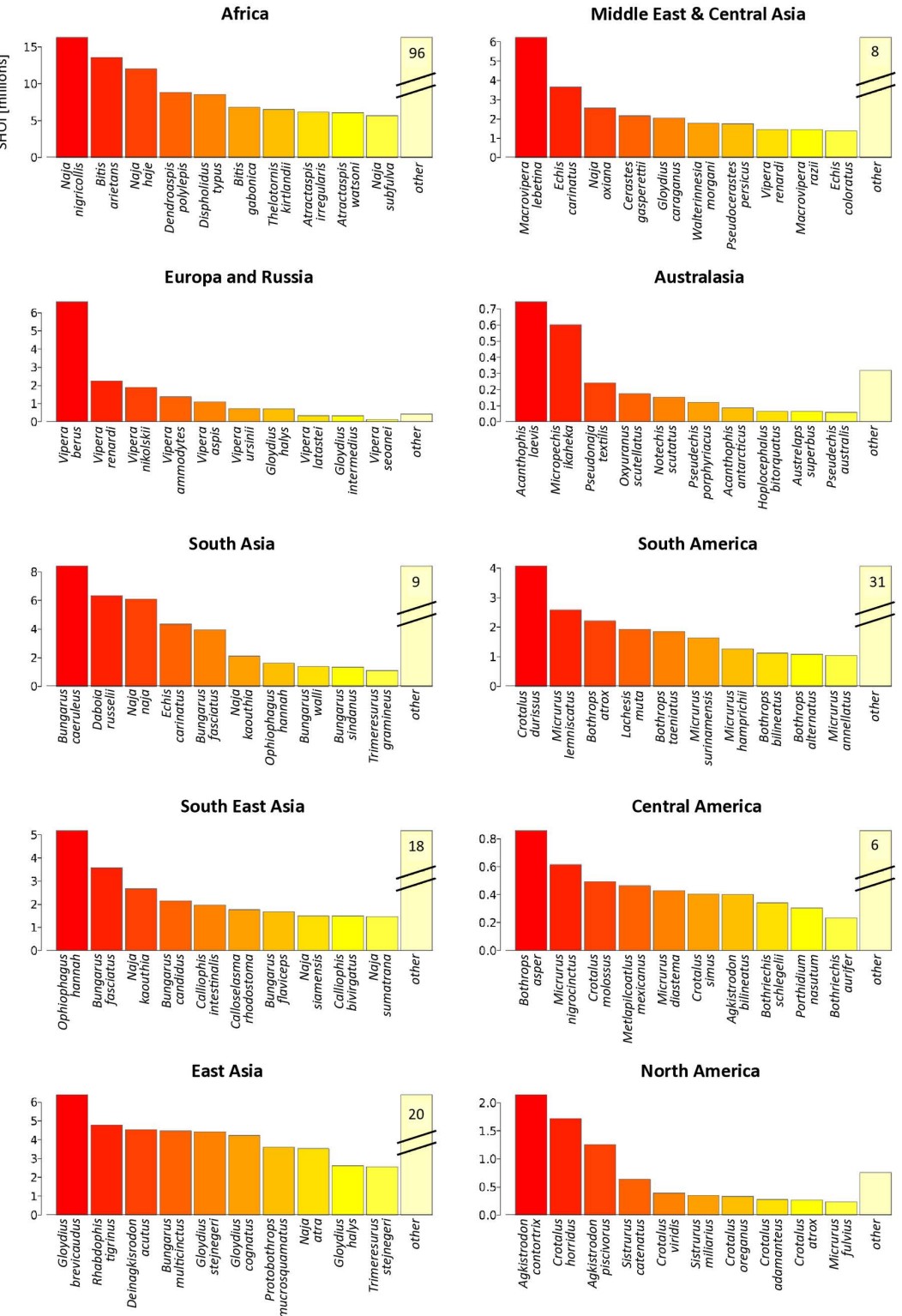

**Fig 9. Most important region-specific snake species based on their SHOI.**

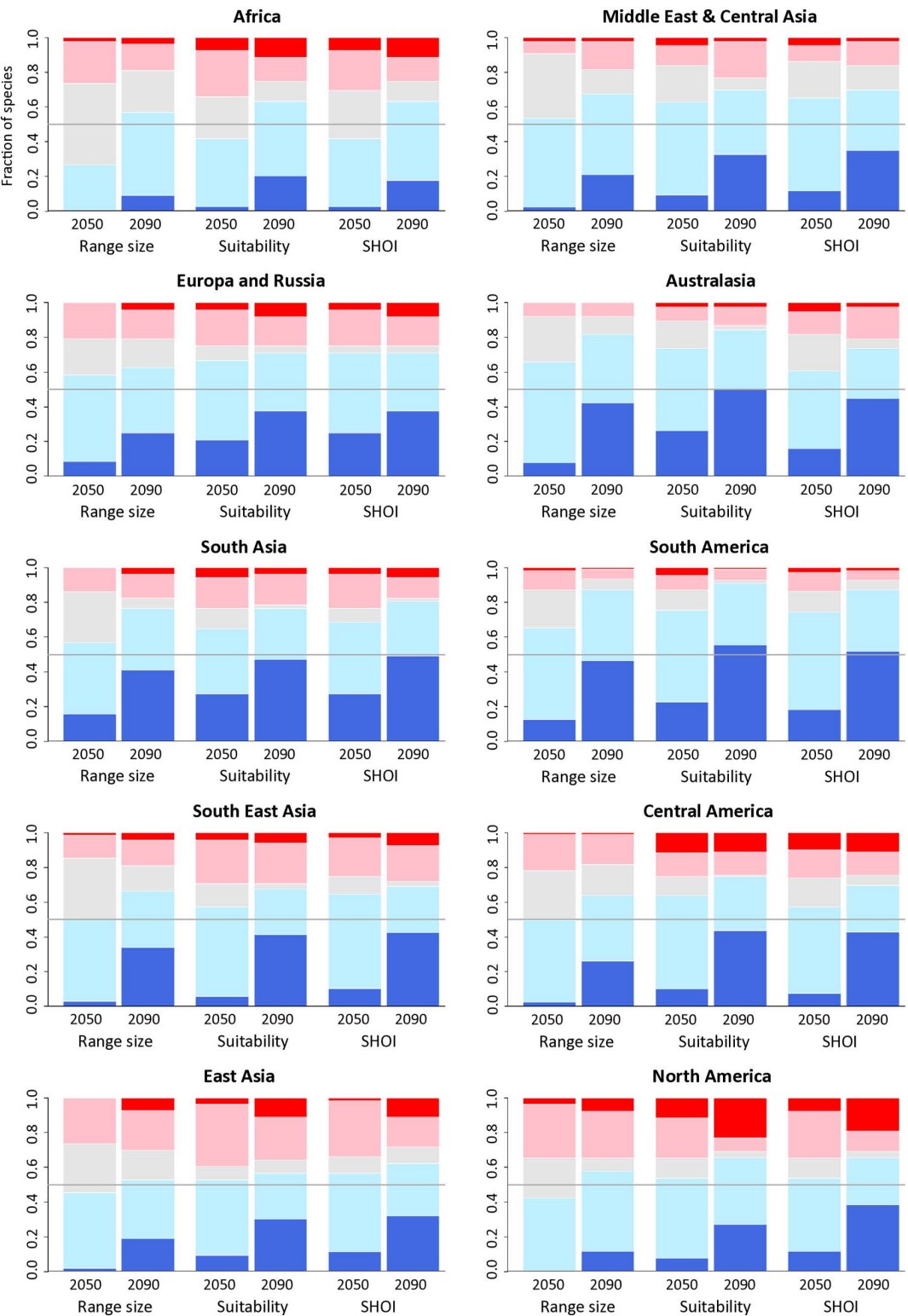

**Fig 10. Predicted increases and decreases in SHOI per region.**

a wide-ranging generalist [30]. For example, *Vipera berus* is by far the most widely spread and data-rich species of MIVS and had a good model fit (test AUC of 0.86) but an average partial ROC of below 1 (0.83), meaning it predicted presences worse than it predicted random points. Similarly, *Bothrops lanceolatus* and *Protobothrops flavoviridis* are island species occupying almost all available habitat surrounded by water and had ROCs not significantly above 1. This does not mean that predictions are not true, but that they are not significantly different from random, i.e., they are probably more limited by available area than by environmental conditions. This shows the benefit of additionally ground-truthing predictions using EDRs. Nevertheless, when using our predictions, more research should be undertaken to verify habitat suitability for the handful of species with poorer model performance. Generally, models for MUs performed well, indicating that the way species were combined into multi-species models if necessary did not result in poorer models. However, we acknowledge that using MUs likely estimates a larger umbrella niche for species complexes than individual species models would and recommend that separate models are run as soon as sufficient data point become available and geographic delineation between lineages are reliably resolved.

Underpredictions of EDRs were low across most species, while overprediction rates were higher for data-poor or geographically restricted species. Importantly, under- and overpredictions should not inherently be conflated with model errors, because EDRs are not 'true' distributions of species. Parts of EDRs that are predicted as unsuitable may be due to patchy suitability within EDRs. In that context distribution models are likely more accurate than EDRs, illustrating one benefit of high resolutions statistical modelling techniques. Additionally, we restricted final models by cost distance to known occurrences. Some parts of EDRs with only anecdotal evidence for species presences may have been cut out because they are in relatively low suitability habitat far from reliable observations. Examples of this are the reported low-density presence of the Black Mamba (*Dendroaspis polylepis*) throughout low suitability habitat in Nigeria, or areas of South Sudan and Sudan, which have sparse observation records for man species due to geopolitical complexities. Such areas need further validation.

Similarly, overpredicted areas compared to EDRs are typically of much lower average suitability than areas within EDRs. Such low suitability may represent low density parts of a snakes' habitat where it is hard to detect, especially since detection rates for snakes are notoriously low at lower population densities and associated low sampling effort [69]. Many predicted suitable areas outside EDRs may, consequently, be the true extents of species' distributions but represent low population densities or sink habitats with occasional occupation. A benefit of combining ENMs with EDRs is the knowledge gained on poorly understood snake distributions [18] and, potentially, abundance patterns [70]. However, targeted ground-truthing of predictions is required because biotic interactions or unknown environmental variables may prevent snakes from occupying otherwise suitable habitat. For example, theoretically suitable habitat for adjacent, closely related species may overlap but one species may outcompete the other [48]. Such is likely the case for the spitting cobras *Naja nigricollis* and *Naja pallida* in East Africa, where our models overpredict *Naja nigricollis* habitat (Stephen Spawls, pers. com.).

The ten species with the highest SHOI in each region represent known patterns of snake importance in some cases but not others. For example, the 'big four' MIVS that present the highest burden of snakebite in India (*Bungarus caeruleus*, *Daboia russelii*, *Naja naja*, and *Echis carinatus*) [71] were also the four species with the highest SHOI in South Asia. In Australasia however, Papua New Guinean (PNG) species *Acanthophis laevis* and *Micropechis ikaheka* dominated the SHOI ranking, whereas most snakebites are caused by taipans (*Oxyuranus scutellatus*) [72], which ranked fourth after the Australian eastern brown snake (*Pseudonaja textiles*), the most medically important snake in Australia [73]. *Micropechis ikaheka* does overlap in its large range with many human populations in PNG but is known to be crepuscular and has low population densities, meaning that despite their range overlap with humans, their encounter frequency is low (David Williams, pers. com.). The pitvipers *Crotalus durissus* and *Bothrops atrox, which cause* large proportions of snakebites in South America [74], were among the 3 highest SHOI species of that region, similar to the number one SHOI species *Bothrops asper* in Central America [74].

In Africa, the top two species *Naja nigricollis* and *Bitis arietans* are known to cause large numbers of bites. However, due to the highly variable African landscape, some groups of species don't necessarily rank highly when visualising the entire continent. For example, none of the carpet vipers (*Echis* spp.) individually rank high across Africa despite their enormous contribution to snakebite number where they are endemic [60,75]. However, when taken as a group, their SHOI ranks second behind black-necked spitting cobras (*Naja nigricollis*). When looking at specific countries, even individual *Echis* spp. rank highly: *Echis romani* ranks second in Nigeria and *Echis leucogaster* ranks first in Niger. For this reason, we supply SHOI values per species as well as aggregated per taxonomic groups, for biogeographic regions and individual countries S1 Appendix.

Changes in habitat suitability and SHOI varied hugely among species and geographically within species (Figs 3 and 4). For example, the wide-ranging species *Bitis arietans* is predicted to experience large decreases in suitability across most of its range, while *Naja nigricollis* is predicted to experience increases. A third species in the same region, *Dendroaspis polylepis* (Fig 3), is predicted to contract its' range in the north but expand it towards the South, particularly along the South African Coast. This highlights the importance of high-resolution analyses and species-specific assessments instead of assuming generalized patterns.

The effects of climate change on species distributions also varied among GCMs and the resulting uncertainty of predictions varied substantially across species' ranges (Fig 3D & 3G). Generally, the uncertainty of future suitability increased from 2050 to 2090 (Fig 5), as expected due to increased uncertainty of climate predictions further into the future. Even when uncertainty in overall changes was high, uncertainty was not necessarily high across the whole range of a species but rather concentrated in some areas (Fig 3). For some species, median changes in SHOI showed clear positive or negative trends, but 80% uncertainty intervals included potential future outcomes where directions of change were opposite to median predictions (e.g., 10% quantile changes were negative while median and 90% quantile changes were positive; see S2 Appendix and S3 Appendix). The species with the highest 'certain' (uniform direction of change across 10%, 90% and median) increase in SHOI were typically wide-ranging species that exhibited a unidirectional range shift, and increased range size and suitability at higher latitudes (Fig 5F, 5I, and 5L). North American species were prevalent in the list of species with high increases in SHOI. More complex patterns in SHOI change were usually seen in species with ranges that straddled the equator (Figs 3 and 5C). Importantly, species that showed less 'certain' average changes in total SHOI nevertheless changed ranges, exposing substantial numbers of 'new' human populations to increased risk of snakebite. A low total change in SHOI should, therefore, not be seen as inconsequential to the distribution of future potential snakebite risk, especially in regions with more complex spatial redistributions of species and high snakebite burden such as Africa, South and South-East Asia. Detailed SHOI change predictions for individual countries are provided in S4 Appendix and should be used by policy makers to guide long term decisions on antivenom procurement and placement.

When looking only at median changes, most species are expected to decrease their range size (Fig 8G), suitability, and SHOI (Fig 10), especially in South-East Asia, the Congo and the Amazon basins (Fig 8A and 8G). This highlights the need for a One-Health approach [76] in simultaneously managing snake conservation with snake-human conflict. Snakes are integral to species rich ecosystems in the tropics [6], which are where snakebite is the most prevalent [5] and where species are most data-sparse [15]. Many MIVS populations are already experiencing pressure from land use change [9] and persecution [12]. Climate change will add to these pressures and potentially have a high impact on range-restricted species or those with patchy distributions [77]. Future anthropogenic land use change and climate change driven vegetation changes will likely further impact snake distributions, while human population growth may increase SHOI further. While we only examine the impact of climate change in our study, these additional drivers of SHOI should be examined further in future studies.

On the other hand, many regions will see an increase in SHOI for several MIVS by 2090 (Fig 10; Africa, Central America, East Asia, and North America). In many cases SHOI increases coincide with expansions of species into new areas (Fig 8D). Such expansions assume, of course, that species can track changes in habitat suitability. We tried to account for

dispersal limitations by restricting the maximum cost-distance from known observations. However, snake species differ in their dispersal capabilities [77] and sedentary species are less likely to realize their full future potential distributions. Nevertheless, the combination of habitat loss for some and increased SHOI for other species highlights the importance of promoting successful co-existence of humans and snakes for conservation and health purposes now more than ever. Community education, snakebite avoidance training, and access to medical treatment are crucial components of a successful strategy to future-proof snakebite mitigation.

The predicted changes in distributions will be important for assessing strategies for adaptation. Treatments for snakebites are based on delivery of antivenoms that are specific to species or genera of snakes (monovalent), or to regions of the world (polyvalent) [17]. Predicting changed distributions and SHOI enables more accurate delivery of the correct treatments and doses to the most appropriate locations, depending on which snakes are causing greatest injuries near a particular hospital or clinic.

We recommend future studies to further build on the dataset we collated, to enable individual models for some of the more data deficient species, and to fill survey gaps and verify our predictions in poorly sampled regions. We hope our data collection and models will present an opportunity for the broader research community to combine efforts and improve our understanding of the spatial distribution of snakebite risk, by incorporating drivers of snake-human conflict in areas of co-occurrence. Furthermore, as high computing power becomes more accessible, we encourage further research into snake distributions using a broader variety of model approaches (e.g., ensemble models), and other approaches that are computationally demanding and were outside of the scope of our study.

Our study highlights trends in snake-human conflict predicted to arise from climate change at an unprecedented resolution. The database of occurrence records, EDRs, distribution models, and taxonomy are vetted and updated at regular intervals in collaboration with experts from around the World, meaning that outputs are not static but adapted to new information as it becomes available. All outputs are freely available to other researchers, NGOs, and governments through OPHIDS, thereby facilitating cross-disciplinary collaboration on snakebite management and mitigation. Some data is also presented as part of a web-portal accessible to the general public to promote education [27]. We hope that these data will present a 'one-stop-shop' for all stakeholders involved in snakebite mitigation, by maximizing collaboration on the development of this, as well as other, derived public goods without duplication of effort and resources.

## Supporting information

**S1 Text. Detailed methodology for data collection and analysis.**
(DOCX)

**S1 Table. Complete list of all species, MUs, and species-specific results for range size, suitability, and SHOI.**
(XLSX)

**S2 Text. Detailed methodology for georeferencing descriptive localities using google maps.**
(DOCX)

**S3 Text. Detailed methodology for environmental layer sourcing and processing.**
(DOCX)

**S1 Appendix. Directory containing graphs of SHOI rankings for species and species groups for biogeographic regions and individual countries.**
(ZIP)

**S2 Appendix. Directory containing graphs of trends in species' range size, suitability, and SHOI by 2050 & 2090 (species names starting with A-L).**
(ZIP)

**S3 Appendix. Directory containing graphs of trends in species' range size, suitability, and SHOI by 2050 & 2090 (species names starting with M-Z).**
(ZIP)

**S4 Appendix. Directory containing graphs of predicted future median SHOI, and changes in SHOI for species for individual countries.**
(ZIP)

## Acknowledgments

We would like to thank our expert panel for their time and expertise and the WHO GIS Centre for Health for their continued contribution to this project, especially Carlos R Ochoa, Kshitij Bhatt, Julia E Coronel, Daniel K Obare, and Ravi S Gopala Krishnan. Data was contributed and vetted by an expert panel that included co-authors of this manuscript and additional experts. Major contributors were Gerardo Martín, Wolfgang Wüster, Romulus Whitaker, Fernando Martínez-Freiría, Kate Jackson, Laurent Chirio, Matthew LeBreton, Abubakr Mohammad Abdalhalee, Deb P Pandey, Chifundera K Zacharie, Maria Elena Barragan-Paladines, Carlos Yañez Arenas, Masoud Yousefi, Patrick K. Malonza, Mahmood Sasa, Zuhair S Amr, Hammadi Achour, Anooshe Kafash, David J Williams, Jordi Tena-Garcés, Kyle B Ray, Stephen Spawls, Jean-Philippe Chippaux, Jean-Francois Trape, Anita Malhotra, Tomas Mazuch, Mark O'Shea, Paola Carrasco, Rafe M Brown, Lise Bethy Mavoungou, Johannes Els, Rushen Bilgin, Rune Midtgaard, Johnathan Lucas, Gnaneswar Ch, Johan Marais, and Sebastián Estrada Gómez among others.

## Author contributions

**Conceptualization:** Anna F. V. Pintor, Mike Turner, Bernadette Abela, David J. Williams.

**Data curation:** Anna F. V. Pintor, Kt Friar, Adam McKay, Gerardo Martín, Wolfgang Wüster, Romulus Whitaker, Fernando Martínez-Freiría, Kate Jackson, Laurent Chirio, Matthew LeBreton, Abubakr Mohammad Abdalhalee, Ulrich Kuch, Deb P. Pandey, Chifundera K. Zacharie, Maria Elena Barragan-Paladines, Carlos Yañez Arenas, Masoud Yousefi, Patrick K. Malonza, Mahmood Sasa, Zuhair S. Amr, Hammadi Achour, Anooshe Kafash.

**Formal analysis:** Anna F. V. Pintor, Kaushi S. T. Kanankege, Adam McKay, Gerardo Martín.

**Funding acquisition:** Bernadette Abela, David J. Williams.

**Investigation:** Anna F. V. Pintor, Kaushi S. T. Kanankege, Mike Turner, Rafael Ruiz de Castañeda, Bethany Moos, Tamer A Hasanein, Prashant Hedao, Kt Friar, David J. Williams.

**Methodology:** Anna F. V. Pintor, Kaushi S. T. Kanankege, Mike Turner, Gerardo Martín, Fernando Martínez-Freiría, David J. Williams.

**Project administration:** Bernadette Abela, Tamer A Hasanein, Prashant Hedao, David J. Williams.

**Resources:** Bernadette Abela, Kt Friar, Adam McKay, Wolfgang Wüster, Romulus Whitaker, Kate Jackson, Laurent Chirio, Matthew LeBreton, Abubakr Mohammad Abdalhalee, Ulrich Kuch, Deb P. Pandey, Chifundera K. Zacharie, Maria Elena Barragan-Paladines, Carlos Yañez Arenas, Masoud Yousefi, Patrick K. Malonza, Mahmood Sasa, Zuhair S. Amr, Hammadi Achour, Anooshe Kafash.

**Supervision:** Bernadette Abela, Mike Turner, Rafael Ruiz de Castañeda, David J. Williams.

**Visualization:** Anna F. V. Pintor.

**Writing – original draft:** Anna F. V. Pintor.

**Writing – review & editing:** Anna F. V. Pintor, Kaushi S. T. Kanankege, Mike Turner, Bernadette Abela, Rafael Ruiz de Castañeda, Bethany Moos, Tamer A Hasanein, Prashant Hedao, Gerardo Martín, Wolfgang Wüster, Romulus Whitaker,

Fernando Martínez-Freiría, Kate Jackson, Laurent Chirio, Matthew LeBreton, Abubakr Mohammad Abdalhalee, Ulrich Kuch, Deb P. Pandey, Chifundera K. Zacharie, Maria Elena Barragan-Paladines, Carlos Yañez Arenas, Masoud Yousefi, Patrick K. Malonza, Mahmood Sasa, Zuhair S. Amr, Hammadi Achour, Anooshe Kafash, David J. Williams.

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
