## [Decision Letter · Decision Letter 0]

31 Jul 2025

Response to Reviewers
Revised Manuscript with Track Changes
Manuscript

Shaden Kamhawi

co-Editor-in-Chief

Paul Brindley

co-Editor-in-Chief

**Journal Requirements:**

At this stage, the following Authors/Authors require contributions: Anna Pintor, Kaushi ST Kanankege, Mike Turner, Rafael Ruiz de Castañeda, Bethany Moos, Tamer A Hasanein, Prashant Hedao, Kt Friar, Adam McKay, Gerardo Martín, Wolfgang Wüster, Romulus Whitaker, Fernando Martínez-Freiría, Kate Jackson, Laurent Chirio, Matthew LeBreton, Abubakr Mohammad Abdalhalee, Ulrich Kuch, Deb P Pandey, Chifundera K Zacharie, Maria Elena Barragan-Paladines, Carlos Yañez Arenas, Masoud Yousefi, Patrick K Malonza, Mahmood Sasa, Zuhair S Amr, Hammadi Achour, Anooshe Kafash, and David J Williams. Please ensure that the full contributions of each author are acknowledged in the "Add/Edit/Remove Authors" section of our submission form.

3) Some material included in your submission may be copyrighted. According to PLOSu2019s copyright policy, authors who use figures or other material (e.g., graphics, clipart, maps) from another author or copyright holder must demonstrate or obtain permission to publish this material under the Creative Commons Attribution 4.0 International (CC BY 4.0) License used by PLOS journals. Please closely review the details of PLOSu2019s copyright requirements here: PLOS Licenses and Copyright. If you need to request permissions from a copyright holder, you may use PLOS's Copyright Content Permission form.

Potential Copyright Issues:

i) Figures 3-8, Sup. Mat. 1, and Sup. Mat. 3. Please (a) provide a direct link to the base layer of the map (i.e., the country or region border shape) and ensure this is also included in the figure legend; and (b) provide a link to the terms of use / license information for the base layer image or shapefile. We cannot publish proprietary or copyrighted maps (e.g. Google Maps, Mapquest) and the terms of use for your map base layer must be compatible with our CC BY 4.0 license.

4) Thank you for uploading your study's underlying data set. Unfortunately, the repository you have noted in your Data Availability statement does not qualify as an acceptable data repository according to PLOS's standards. At this time, please upload the minimal data set necessary to replicate your study's findings to a stable, public repository (such as figshare or Dryad) and provide us with the relevant URLs, DOIs, or accession numbers that may be used to access these data. We notice that there is a Thank you for uploading your study's underlying data set. Unfortunately, the repository you have noted in your Data Availability statement does not qualify as an acceptable data repository according to PLOS's standards.    At this time, please upload the minimal data set necessary to replicate your study's findings to a stable, public repository (such as figshare or Dryad) and provide us with the relevant URLs, DOIs, or accession numbers that may be used to access these data. For a list of recommended repositories and additional information on PLOS standards for data deposition, please see license on your data. We would encourage you to consider using a license that is no more restrictive than CC BY, in line with PLOS’ recommendation on licensing (http://journals.plos.org/plosone/s/licenses-and-copyright). For a list of recommended repositories and additional information on PLOS standards for data deposition, please see https://journals.plos.org/plosntds/s/recommended-repositories

**Reviewers' comments:**

**Key Review Criteria Required for Acceptance?**

**Methods**

-Are the objectives of the study clearly articulated with a clear testable hypothesis stated?

-Is the study design appropriate to address the stated objectives?

-Is the population clearly described and appropriate for the hypothesis being tested?

-Is the sample size sufficient to ensure adequate power to address the hypothesis being tested?

-Were correct statistical analysis used to support conclusions?

-Are there concerns about ethical or regulatory requirements being met?

Reviewer #1: As a positive aspect, the authors analyze a large number of species. I consider it important to distinguish between Type I and Type II species in the analyses. It is well known that Type II species have the potential to cause severe envenomations but rarely do so. How many of the species included are Type I and how many are Type II? I raise this point because a high proportion of Type II species may lead to an overestimation of the problem in certain regions. If most of the species included in the study are Type II, the conclusions may not be directly related to actual snakebite incidents.

Another issue is that the authors state they included several species, but in reality, many of them were modeled as a single entity (modeling units – MUs), and only after modeling the species complexes together did they separate the species (lines 268–279). This approach assumes that all lineages within a species complex share the same climatic niche. However, this is not necessarily true, especially in cases where the complex is distributed along an environmental gradient. Each lineage may respond differently to climate change, and modeling them together implies assuming a single niche for all of them. Additionally, this leads to an overestimation of the number of species modeled, since the actual number is lower due to several species being grouped as one. This is particularly relevant because, in principle, the inclusion of many species is presented as one of the strengths of the manuscript. I consider this approach flawed, and the authors should be fully aware of the limitations of modeling using MUs.

Along these lines, the authors give the example of the inclusion of the five species formerly classified as Naja melanoleuca. In this case, it is unclear whether they were modeled as a complex or separately. If modeled as a complex, it would be similar to modeling a single species (as in previous studies), and if modeled separately, I have some questions. Reviewing Wüster et al. (2018), which describes these five species, it is stated, for example, that Naja melanoleuca and Naja subfulva occur sympatrically in some areas and are difficult to distinguish morphologically:

"Some of these specimens and populations are difficult to differentiate from sympatric N. melanoleuca, necessitating the use of DNA barcoding approaches (Hebert et al., 2003; Clause et al., 2016) to ensure reliable identification."

How did the authors distinguish occurrence points in those locations, especially when using GBIF data? Furthermore, one of the species, Naja peroescobari, is found on an island—how was this species modeled?

Another relevant issue is the excessive reliance on expert opinion throughout the manuscript. Ideally, information used in a scientific paper should come from published sources, not primarily from expert panels. Some examples include:

Line 140: “Data collation included rigorous vetting by an international expert panel comprised of leading researchers in the fields of snake distributions and taxonomy to guarantee a trustworthy one-stop-shop for researchers, governments, NGOs, and the general public.”

Line 178: “All occurrence records were vetted for taxonomic and location accuracy by an expert panel of >30 experts from around the world.”

Line 215: “Only two sets of collinear variables were maintained based on expert experience of complementarity of those variables in other analyses despite collinearity.” This is problematic—despite acknowledging serious collinearity issues, the authors retained collinear variables based solely on expert opinion.

Line 403: “Underpredictions were common where anecdotal expert knowledge suggests occasional presences without accurate location information.”

Line 483: “We established a panel of experts to vet all data and outputs, lending additional reliability to our results.”

My next major concern involves the construction of the species distribution models. The implementation lacks sufficient detail to assess its rigor. Specific concerns include:

(i) The authors do not specify how many background points were used per species.

(ii) It is unclear how the test dataset was selected—I assume a random 30% split. If so, this is suboptimal due to strong spatial autocorrelation. A geographically structured cross-validation approach (e.g., Muscarella et al., 2014; https://doi.org/10.1111/2041-210X.12261) should be used to reduce spatial bias.

(iii) The authors should filter out points with high climatic autocorrelation.

(iv) It is not clear how the authors selected among linear, quadratic, product, or hinge feature types. Was it based on AIC? AUC?

(v) The rm hyperparameter in Maxent should be evaluated.

(vi) Variable elimination was based on permutation importance, but if there were only 20 presence points and 10,000 background points (exact numbers are not given), removing 15 predictors does not produce a balanced model.

(vii) Some variables were kept static even though they are expected to change in the future. How does this impact projections? Would using only climatic variables yield different results? A sensitivity analysis is needed.

(viii) Line 306: How was the cost of 500 determined? Based on what criteria? Many decisions seem arbitrary and lack clear justification.

(ix) More than one climate scenario should be included—standard practice involves using at least three (RCP2.6, RCP4.5, RCP8.5).

(x) Why was Maxent selected? While I do not object to its use, a stronger justification is required. Why not use ensemble models?

My final methodological concern relates to the use of human population density. While its inclusion is conceptually interesting, the authors fail to distinguish which segments of the population are actually at risk (e.g., rural workers, rural populations). Large metropolitan areas may have high habitat suitability for snakes and high human density, but still experience low incidence of snakebite. Moreover, the study assumes a static human population density for future projections, which is unrealistic. Future human density models should be incorporated into the projections.

Lines 331–334: The study by Martin et al. is well conducted and finds an association between estimated abundance and incidence. However, their approach is based on Point Process Models, which differ significantly from the method used in this article. In contrast, the work by Yañez-Arenas et al. (2016) does find an association between the sum of suitabilities (binary maps) and snakebite incidents, although only in certain countries. Therefore, I believe it would be beneficial to incorporate an analysis that can more robustly support the validity of that claim. Once again, including only Type I species in this estimate could be helpful.

Reviewer #2: The paper is very well written and meets the review criteria needed for publication.

Reviewer #3: The manuscript articulates clear objectives, namely to improve global understanding of the distribution of medically important venomous snakes (MIVS) and to model the overlap between snake distributions and human populations under climate change. The hypothesis — that climate change will shift snake ranges and alter human exposure — is well grounded and testable.

The study design is appropriate and innovative, utilizing expert-derived ranges (EDRs), extensive occurrence records, and environmental niche models (ENMs) at high spatial resolution (~1 km). The iterative process for taxonomy validation and model vetting adds rigor and credibility.

Key strengths:

- Extensive dataset covering 508 MIVS and 314 modeling units (MUs).

- Expert vetting of occurrence data and spatial layers.

- Use of robust modeling techniques (Maxent, cost distance thresholds)

- Application of a "One Health" lens.

Concerns:

- While the choice of Maxent is justified due to computational limits, lack of ensemble modeling may affect robustness in areas of ecological uncertainty. The authors should acknowledge this trade-off more clearly.

- Inclusion of only one SSP pathway (SSP5-8.5) for climate change limits policy utility; a short justification is offered, but considering recent shifts in emissions policy, it would be beneficial to clarify limitations further.

- The cutoff of <20 occurrence records per species/MU for full modeling is reasonable but could be supplemented by uncertainty metrics or maps to explicitly convey model confidence.

The study uses secondary, non-human data, meaning there are no ethical concerns regarding subject use or data acquisition.

**Results**

-Does the analysis presented match the analysis plan?

-Are the results clearly and completely presented?

-Are the figures (Tables, Images) of sufficient quality for clarity?

Reviewer #1: Overall, the results are presented appropriately, and the figures are of good quality.

The authors should avoid including references in the results section; this section should be rewritten to prevent mixing results with the discussion.

Reviewer #2: The results are presented clearly and are based on the analysis that was conducted. The figures are of high quality and enable easy interpretation of the data presented within the results section.

Reviewer #3: The results are thoroughly presented and supported by high-quality figures (e.g., maps of current and future species distributions, Snake-Human Overlap Index). The results align with the methods and hypotheses and present a convincing case for shifts in both biodiversity and human exposure.

Noteworthy aspects incude:

- Over 90% of ENMs had AUC > 0.85, indicating strong predictive performance.

- Clear documentation of which species are expected to see range contractions vs. expansions.

- Stratification of results by biogeographic region adds policy relevance.

- Detailed analysis of agreement/disagreement between ENMs and EDRs demonstrates transparency and robustness.

Minor issues:

- Figures are high quality but sometimes dense; perhaps include regional insets or simplify overlays for readability.

- A table summarizing species with highest projected increase in SHOI per region would enhance usability by health policymakers.

**Conclusions**

-Are the conclusions supported by the data presented?

-Are the limitations of analysis clearly described?

-Do the authors discuss how these data can be helpful to advance our understanding of the topic under study?

-Is public health relevance addressed?

Reviewer #1: One concern is that many of the conclusions are not new. For example, the main conclusion in the abstract states: “We predict substantial, short- and long-term shifts in snake distributions, including range contractions for many threatened species and increased human exposure to species of major public health concern.” However, upon reviewing Martines et al. (2024), their abstract presents a very similar conclusion: “Losses of potentially suitable areas for the survival of most venomous snake species will occur by 2070. However, some species of high risk to public health could gain climatically suitable areas for habitation.”

This overlap is also evident in the discussion, where the authors highlight key snake species responsible for envenomations (e.g., Bungarus caeruleus, Daboia russelii, Naja naja, Echis carinatus, Crotalus durissus, Bothrops atrox, Bothrops asper, Naja nigricollis, Bitis arietans, among others), most of which were already included in Martines et al. (2024). Similarly, the conclusion that snake distributions will contract in the Congo and Amazon rainforests is also consistent with the findings of Martines et al. (2024).

I believe the manuscript would benefit greatly if the authors clearly emphasize what the inclusion of new data contributes beyond what has already been demonstrated in previous studies.

One possible improvement would be to incorporate the adaptive capacity of species to climate change, as demonstrated by Diniz-Filho et al. (2019) (https://doi.org/10.1111/ecog.04264) and Souza et al. (2023) (10.3389/fevo.2023.1038018). This addition would strengthen the manuscript by providing a more accurate assessment of whether species projected to lose habitat will actually decline or potentially adapt to new conditions. Simply increasing the number of species included may not constitute sufficient novelty for a high-impact journal such as PNTD.

Reviewer #2: The conclusions drawn from the data are clearly thought through. Potential limitations are discussed and incorporated into the explanation while addressing the potential impact of the work from a public health perspective.

Reviewer #3: The conclusions are strongly supported by the data and analysis. The authors correctly highlight the dual implications for public health and conservation. They do not overstate their findings and are appropriately cautious regarding uncertainties.

Highlights include:

- Emphasis on the dynamic and updating nature of the OPHIDS database is timely and commendable.

- Appropriate call for region-specific One Health interventions.

- Clear articulation of how this work supports WHO's 2030 snakebite goals.

Nevertheless, the following are suggested improvements:

- Better distinguish between "exposure risk" (overlap) and actual bite risk, which includes behavioral and cultural factors.

- More explicitly state how governments or researchers can use these outputs — e.g., in hospital planning or antivenom supply chains.

**Editorial and Data Presentation Modifications?**

Reviewer #1: (No Response)

Reviewer #2: I have no suggestions for data modifications.

Reviewer #3: The following edtorial modifications are suggested:

- Improve clarity of the abstract. Currently too dense and jargon-heavy; streamline and define SHOI.

- Minor typographical and grammatical inconsistencies throughout (e.g., spacing, abbreviations).

- Consider including a glossary of terms (SHOI, ENM, MU) for broader accessibility.

- Expand legends and titles in some figures to clarify differences between present and future projections.

**Summary and General Comments**

Reviewer #1: The manuscript titled "Climate change induced complex shifts in snake distributions expose people to snakebite and threaten biodiversity" addresses an important topic: the effect of climate change on the distribution of venomous snake species. I had previously reviewed this manuscript for another journal, and I am sincerely disappointed that the authors have not incorporated any of the suggestions I made during that earlier review. Therefore, my current evaluation remains the same as before.

The paper deals with an interesting subject: the effect of climate change on the distribution of snake species that pose a risk to humans. The strongest aspect of the study is the inclusion of a larger number of species compared to previous studies. However, from a methodological perspective, the work presents some limitations, and its conclusions are constrained.

Reviewer #2: I felt this paper was very well written and I have no concerns to raise after reading through it. The work presented here could be used to help guide stakeholders involved in the prevention of snakebite (as discussed).

Reviewer #3: Strengths:

- Unprecedented global scope and resolution for MIVS modeling.

- Integration of updated taxonomy and vetted occurrence data.

- Methodologically robust ENMs and well-articulated analytical pipeline.

- High policy relevance for climate adaptation, conservation, and snakebite management.

Weaknesses:

- Reliance on a single climate scenario limits forward-looking adaptability.

- Assumes dispersal without addressing species-specific barriers in detail.

- Human behavioral variability (e.g., use of protective equipment, awareness) not integrated into SHOI or modeling.

- Novelty and Significance:

- Highly original contribution with direct implications for WHO targets.

- Sets a new benchmark for predictive modeling in the NTD and biodiversity domains.

Suggested Revisions:

- Provide a justification for the lack of ensemble modeling or include a brief sensitivity analysis where feasible.

- Discuss limitations of projecting only SSP5-8.5 in more detail and its implications for planning.

- Improve abstract readability and figure clarity.

- Add a concise user guide or example on how policymakers or local health agencies can use the database.

PLOS authors have the option to publish the peer review history of their article (what does this mean? ). If published, this will include your full peer review and any attached files.

**Do you want your identity to be public for this peer review?** For information about this choice, including consent withdrawal, please see our Privacy Policy .

Reviewer #1: No

Reviewer #2: No

Reviewer #3: **Yes:** Wisdom Mdumiseni D. Dlamini

**Figure resubmission:****Reproducibility:** To enhance the reproducibility of your results, we recommend that authors of applicable studies deposit laboratory protocols in protocols.io, where a protocol can be assigned its own identifier (DOI) such that it can be cited independently in the future. Additionally, PLOS ONE offers an option to publish peer-reviewed clinical study protocols. Read more information on sharing protocols at https://plos.org/protocols?utm_medium=editorial-email&utm_source=authorletters&utm_campaign=protocols

---

## [Decision Letter · Decision Letter 1]

11 Dec 2025

Climate change induced complex shifts in snake distributions expose people to snakebite and threaten biodiversity

Dear Dr. Pintor,

Thank you for submitting your manuscript to PLOS Neglected Tropical Diseases. After careful consideration, we feel that it has merit but does not fully meet PLOS Neglected Tropical Diseases's publication criteria as it currently stands. Therefore, we invite you to submit a revised version of the manuscript that addresses the points raised during the review process.

Please submit your revised manuscript within by Jan 10 2026 11:59PM. If you will need more time than this to complete your revisions, please reply to this message or contact the journal office at plosntds@plos.org. Please include the following items when submitting your revised manuscript:

We look forward to receiving your revised manuscript.

Kind regards,

Wuelton Monteiro, Ph.D.

Section Editor

Wuelton Monteiro

Section Editor

Shaden Kamhawi

co-Editor-in-Chief

Paul Brindley

co-Editor-in-Chief

**Journal Requirements:**

**Reviewers' Comments:**

Reviewer's Responses to Questions

**Key Review Criteria Required for Acceptance?**

**Methods**

-Are the objectives of the study clearly articulated with a clear testable hypothesis stated?

-Is the study design appropriate to address the stated objectives?

-Is the population clearly described and appropriate for the hypothesis being tested?

-Is the sample size sufficient to ensure adequate power to address the hypothesis being tested?

-Were correct statistical analysis used to support conclusions?

-Are there concerns about ethical or regulatory requirements being met?

Reviewer #1: First, I would like to thank the authors for addressing my questions and comments on the manuscript. Some of these concerns were resolved through the responses and modifications in the text. However, I believe that some methodological and writing issues still remain.

1- Regarding collinearity among variables.

The authors argue that collinearity does not affect the Maxent model (based on reference 31). This is true, but collinearity can affect the model’s projections in new environments. I provide a passage from the abstract of reference 31:

“… (c) collinearity shift and environmental novelty can negatively affect Maxent model transferability. We therefore recommend to quantify and report collinearity shift and environmental novelty to better infer model accuracy when models are spatially and/or temporally transferred.”

Following what the authors cite, I recommend showing whether collinearity shifts exist in the future. It is not necessary to do this for each species model, but at least for each global bioregion. Another valid option is to model without collinear variables.

2- Regarding the test dataset.

Having an independent test dataset is fundamental to evaluate model performance and determine whether the model can make accurate predictions. I consider that the authors should report the AUC (and at least one additional metric, such as TSS or KAPPA) for the test dataset of each model, and not only for the training dataset. Ideally, the test dataset should be partially independent of the training dataset, meaning it should not be just a random 30% of the records.

3- Use of multiple climate scenarios.

This is practically mandatory in climate projection studies. The authors themselves note that climate change is likely to exceed the expectations of the RCP4.5 scenario; it is more likely to fall between RCP4.5 and RCP8.5. Therefore, I strongly suggest presenting models for both scenarios and using at least two uncorrelated Global Circulation Models, which is the minimum recommended in climate change studies. A valid alternative is to use 2.5 arcmin grid cells, which still allows for a clear representation of the margins of change in distributions between the two scenarios; more detailed information could be included in the supplementary material. For a global study, using 1 km resolution is not necessary. Today, there are cloud computing options that allow these analyses to be performed, so computational capacity does not seem to be a valid limitation.

Reviewer #3: The study has a very clear goal: to map where 508 venomous snakes live now and where they might move in the future due to climate change. The authors also want to see how much these snakes will overlap with human populations (the Snake-Human Overlap Index, or SHOI).

The design is appropriate. They used a computer modeling program called Maxent to predict suitable habitats. Because some snake species are rare and have little data, the authors grouped them into "Modelling Units" (MUs). This was a smart and necessary step to make sure all species were included. To ensure accuracy, they checked their computer models against maps drawn by experts.

The population is clearly described: they included every single medically important snake species listed by the WHO (Category 1 and 2). They addressed previous concerns about Category 2 snakes effectively, noting that these snakes can still be very dangerous but are often just less studied.

A main limitation is that they only looked at one future climate scenario (SSP5-8.5, or "business as usual"). However, they explained that running multiple scenarios for over 300 groups of snakes would require too much computer power. They also assumed human populations stay in the same place (using 2020 data), which is a weakness, but understandable for the scope of this study. There are no ethical concerns since they used existing data.

**Results**

-Does the analysis presented match the analysis plan?

-Are the results clearly and completely presented?

-Are the figures (Tables, Images) of sufficient quality for clarity?

Reviewer #1: (No Response)

Reviewer #3: The results follow the plan exactly. The authors present the data clearly, showing which snakes will see their homes shrink (contractions) and which will expand into new areas.

The analysis is honest. They show that most of their models worked very well (AUC score > 0.85), but they admit that models for very wide-ranging snakes or data-poor snakes were less perfect. This honesty makes the results more trustworthy.

The figures are high quality. Figure 5 is especially helpful because it uses color shading to show "uncertainty"—basically, how sure they are about the future predictions. Figure 7 gives a great quick view of the world, showing that snake-human overlap will likely increase in places like the USA, China, and India. As requested in previous reviews, they moved references out of the Results section to keep it clean.

**Conclusions**

-Are the conclusions supported by the data presented?

-Are the limitations of analysis clearly described?

-Do the authors discuss how these data can be helpful to advance our understanding of the topic under study?

-Is public health relevance addressed?

Reviewer #1: 4- Citation of previous studies.

The authors state in their response:

“We believe that agreement of our predicted trends with the lower resolution predictions of previous studies for species subsets is a positive outcome because it confirms that a more complete analysis including all relevant species verifies the trends and shows that these predictions are robust and reliable.”

If previous studies have already reached similar conclusions, the authors should cite them and discuss how their results align with previous findings. As written, it gives the impression that this study is the first to reach these conclusions, which could be considered a careless acknowledgment of the existing literature, whereas referencing previous studies would strengthen the interpretation and relevance of the results.

Reviewer #3: The conclusions match the data. The authors predict that many snake species will shift their ranges toward the poles (away from the equator) to find cooler temperatures. They correctly identify that this will expose new human populations to snakes they haven't dealt with before.

The authors are careful not to overstate their findings. They explain that just because snakes and humans overlap (SHOI), it doesn't automatically guarantee more bites—behavior and healthcare access matter too.

They do a good job discussing the limitations, such as the fact that they grouped some species into "Modelling Units," which might hide small differences between closely related snakes. They strongly argue that this data is vital for public health, specifically for deciding where to store antivenoms in the future.

**Editorial and Data Presentation Modifications?**

Reviewer #1: (No Response)

Reviewer #3: The paper is well written. The authors added a "Glossary" to explain technical terms like "SHOI" and "MUs," which makes the paper much easier to read for non-experts.

They also created a public database called OPHIDS so other researchers can use their data. This is a great contribution to science. They added new graphs for individual countries in the supplementary material (Sup. Mat. 7), which is very helpful for local policymakers who need specific data for their region. No further major changes are needed.

**Summary and General Comments**

Reviewer #1: First, I would like to thank the authors for addressing my questions and comments on the manuscript. Some of these concerns were resolved through the responses and modifications in the text. However, I believe that some methodological and writing issues still remain.

Reviewer #3: This is a landmark study. Unlike previous papers that only looked at a few snakes or specific regions, this study attempts to model all 508 medically important snakes globally. This fills a massive gap in our knowledge. The only weakness is the use of only one climate scenario ("business as usual") and static human population numbers. While this limits the precision of the 2090 predictions, the authors justified it well based on the massive computing power required to do more

PLOS authors have the option to publish the peer review history of their article (what does this mean? ). If published, this will include your full peer review and any attached files.

**Do you want your identity to be public for this peer review?** For information about this choice, including consent withdrawal, please see our Privacy Policy .

Reviewer #1: No

Reviewer #3: **Yes:** Wisdom Mdumiseni D. Dlamini

**Figure resubmission:**
---

## [Editor Report · Decision Letter 2]

11 Feb 2026

Dear Dr Pintor,

We are pleased to inform you that your manuscript 'Climate change induced complex shifts in snake distributions expose people to snakebite and threaten biodiversity' has been provisionally accepted for publication in PLOS Neglected Tropical Diseases.

Best regards,

Wuelton Monteiro, Ph.D.

Section Editor

Wuelton Monteiro

Section Editor

Shaden Kamhawi

co-Editor-in-Chief

Paul Brindley

co-Editor-in-Chief

---

## [Editor Report · Acceptance letter]

Dear Dr Pintor,

We are delighted to inform you that your manuscript, "Climate change induced complex shifts in snake distributions expose people to snakebite and threaten biodiversity," has been formally accepted for publication in PLOS Neglected Tropical Diseases.

Best regards,

Shaden Kamhawi

co-Editor-in-Chief

Paul Brindley

co-Editor-in-Chief
